# "Are we talking just a bit of water out of bank? Or is it Armageddon?" Front line perspectives on transitioning to probabilistic fluvial flood forecasts in England

Louise Arnal[1,2], Liz Anspoks[3], Susan Manson[3], Jessica Neumann[1], Tim Norton[3], Elisabeth Stephens[1], Louise Wolfenden[3], Hannah Louise Cloke[1,4,5]

[1]University of Reading, UK
[2]European Centre for Medium-Range Weather Forecasts, UK
[3]Environment Agency, UK
[4]Uppsala University, Sweden
[5]Centre of Natural Hazards and Disaster Science, Sweden

*Correspondence to: Louise Arnal (louise.arnal@usask.ca)*

**Abstract.** By showing the uncertainty surrounding a prediction, probabilistic forecasts can give an earlier indication of potential upcoming floods, increasing the amount of time available to prepare. However, making a decision based on probabilistic information is challenging. As part of the UK-wide policy's move towards forecast-based flood risk management, the Environment Agency (EA), responsible for managing risks of flooding in England, is transitioning towards the use of probabilistic fluvial forecasts for flood early warning. While science and decision-making are both individually progressing, there still lacks an ideal framework for the incorporation of new and probabilistic science in decision-making practices, and, respectively, the uptake of decision-makers' perspectives in the design of scientific practice. To address this, interviews were carried out with EA decision-makers (i.e. Duty Officers), key players in the EA's flood warning decision-making process, to understand how they perceive this transition might impact on their decision-making. The interviews highlight the complex landscape in which EA Duty Officers operate and the breadth of factors that inform their decisions, additionally to the forecast. Although EA Duty Officers already account for uncertainty and communicate their confidence in the forecast they currently use, the interviews revealed a decision-making process which is still very binary and linear to an extent, which appears at odds with probabilistic forecasting. Based on the interview results, we make recommendations to support a successful transition to probabilistic forecasting for flood early warning in England. These recommendations include: the inclusion of Duty Officers in the new system's design process, the preparation of clear guidelines on how probabilistic forecast should be used for decision-making in practice, the EA communication with all players in the decision-making chain (internal and external) that this transition will become operational practice and the documentation of this transition to help other institutes yet to face a similar challenge.

We believe that this paper is of wide interest for a range of sectors at the intersection between geoscience and society. A glossary of technical terms is highlighted by asterisks in the text and included in Appendix A.

## 1 Introduction

The ongoing shift in UK policy from 'flood defence' towards a forecast-based 'flood risk* management' approach to better anticipate floods (Dale et al., 2012; McEwen et al., 2012) has shaped a series of developments in the uptake of flood forecasting science in practice, often implemented in the wake of significant flood events. Following the summer 2007 UK floods, the development of the National Flood Forecasting System (NFFS) and the Flood Forecasting Centre (FFC*; a UK Met Office and Environment Agency (EA)* partnership; Defra, 2014) were prioritised, with the aim of improving national flood warning services (Pitt, 2008; Stephens and Cloke, 2014). The winter 2013/14 UK floods illustrated the value of these institutional changes to flood forecasting, as well as the value in using new forecasting techniques, such as ensemble* surge forecasts*, for

flood preparedness* (Flowerdew et al., 2009; Stephens and Cloke, 2014). It was also during the 2013/14 floods that the EA moved from using a single prediction of upcoming floods (known as a deterministic forecast*) to using two fluvial (river) flood scenarios*; a 'Best Estimate'* and a 'Reasonable Worst Case'* (see more information below); for flood incident management in England (FFC, 2017). However, the recent winter 2019/2020 UK floods have shown that this approach could be further improved to better capture the uncertainty* in upcoming floods and communicate risk more effectively:

*"The recent floods exposed a limitation in our forecasting approach by only running flood models with a Best Estimate and Reasonable Worst Case meteorological inputs. Whilst this approach is fine at providing a high level general meteorological input to flood forecast models small variances in rainfall profiles within and across catchments makes a big difference in river response and flood risk. Modelling the impacts of snow accumulation and melting was also a particular problem [...] Being able to run meteorological ensembles through our flood forecasting models to determine*

*the probability of different magnitudes of flood impacts within and across catchments would in my view given us a better understanding of river response and allowed for clearer communication of risk from forecasters to responders. [...] There was a lot we did well but a lot we can do better" (Neil Ryan, lead modelling and forecasting duty officer at the EA Leeds Forecasting Centre, March 2020)*

        In 2016, the UK National Flood Resilience Review (NFRR; HM Government, 2016; House of Commons - Environment, Food

and Rural Affairs Committee, 2016) indeed recommended a better integration of probabilistic forecasts* of the weather into flood forecast products to improve the characterisation of uncertainty in future water levels, and to enhance the communication of flood risk and likelihood for informing a range of flood management measures* (as the quote above alludes to). Probabilistic flood forecasts express the likelihood of possible future high river flow scenarios, and can be produced by forcing* a hydrological model* with an ensemble of future meteorological scenarios (Cloke and Pappenberger, 2009). By indicating how

likely a flood is to occur, probabilistic forecasts communicate an estimate of the uncertainty surrounding a prediction (expressed as a probability), and can support risk-based decision-making through an increased probability of detection of floods (reducing missed events*), an earlier indication of potential future extreme events, such as floods, and their associated impacts (Buizza, 2008; Verkade and Werner, 2011; Dale et al., 2012; Stephens and Cloke, 2014). Based on research for the Thames river basin (UK), New et al. (2007) showed that probabilistic forecasts provide more informative results (i.e. by

allowing to quantify the potential impacts of upcoming floods and their associated likelihood) than a scenario-based approach, as is currently used operationally at the EA.

        The EA, an executive non-departmental public body sponsored by Defra* (the UK government Department for Environment, Food & Rural Affairs), is responsible for the operational management of flood risks from rivers and the sea in England (under the Flood and Water Management Act 2010; Werner et al., 2009; Defra, 2010; 2014; Pilling et al., 2016). Their role is to warn

and inform the public and businesses about impending coastal and fluvial floods. The EA also has a strategic overview role for all sources of flooding and works with lead local flood authorities (i.e. emergency responders category 1 (e.g. police services, fire and rescue authorities) and 2 (e.g. utilities, telecommunications, transport providers, Highways Agency)) by providing guidance, knowledge and support in responding to flooding. The EA 'Monitoring and Forecasting Duty Officers' (MFDOs) and 'Flood Warning Duty Officers' (FWDOs) are two roles at the heart of the EA's internal forecast-led decision-

making process and are responsible for coordinating local flood warning.

        It is within the remit of their responsibilities for the EA to move away from incident response and implement flood risk management policy. As part of this wider move and since 2016, the EA's flood incident management strategy* is based on the principle: "think big, act early, be visible" (EA, 2018a). Below the umbrella of this principle, the EA's objectives are to quantify uncertainty, make decisions around incident preparation and escalation (to ensure that resources are put in place early and that

the EA is prepared to scale-up or -down closer to the potential incident; e.g. expanding incident rotas with Duty Officers on standby, requesting equipment to support preventative and/or repair work, such as temporary barriers and pumps) and

communicate flood risk clearly internally and externally (Tim Norton, personal communication). To this end, based on hydro-meteorological forecasts received from the FFC daily, the EA currently (and since the UK winter floods of 2013/14 as mentioned above) produce two deterministic fluvial flood scenarios with a five-day lead time*, a 'Best Estimate' and a 'Reasonable Worst Case'. The EA's operating practice provides guidance on how to use these scenarios to support decision-making for a range of flood incident management activities, in line with the EA's principle. In summary, the 'Reasonable Worst Case' gives an indication of what 'could' happen (i.e. the upper range of forecast rainfall, river conditions and impacts that may occur) and should be used for preparation and informing others. The 'Best Estimate' gives an indication of what 'should' happen (i.e. the middle range of forecast rainfall, river conditions and impacts that may occur) and should be used as the basis for planning where and when to issue flood warnings (EA, 2018b). Together, the two scenarios provide the scale and size of the incident for planning and response preparations (FFC, 2017).

The two EA flood scenarios are an intermediate step between deterministic and probabilistic fluvial flood forecasting (n.b. probabilistic coastal flood forecasts are already operational and this paper focuses on fluvial flood forecasting), as outlined as essential by the NFRR. The new probabilistic flood forecasting system is currently being technically developed and several feasibility projects have been carried out to guide the FFC's move towards probabilistic fluvial flood forecasts (Pitt, 2008; Orr and Twigger-Ross, 2009; Sene et al., 2007; 2009; 2010; Dale et al., 2013). However, as the EA will have to accommodate these probabilistic fluvial flood forecasts in practice, there is still a lack of clarity about how they should be used for flood incident management and by the EA Duty Officers' for flood warning decision-making.

Probabilistic forecasts can be challenging to use for operational decision-making*, given the explicit uncertainty information they communicate (Nicholls, 1999; Cloke and Pappenberger, 2009; Demeritt et al., 2010; Nobert et al., 2010; Ramos et al., 2010; Stephens et al., 2012). Having to translate a range of possible outcomes into an operational decision (such as sending out a flood warning) is intricate and requires careful interpretation, and an understanding of probabilities, risk, uncertainty and of the systems modelled (Dessai and Hulme, 2004). Furthermore, warning based on low probabilities of a flood, for example, will reduce the chance of missing an event, but might also lead to more false alarms*. Decisions can be made following a set of rules, such as threshold exceedance (Dale et al., 2013). However, the decision-making process is complex and generally influenced by several additional factors. These include: the event type (e.g. a localised small flood event vs a large scale extreme flood event), the costs of taking action vs not taking action, the decision-maker's experience of past events, trust in the forecast (which can be built up over time), personal risk aversion, and the cultural context in which decisions are made (Cloke et al., 2009; Arnal et al., 2016; Neumann et al., 2018).

The aim of this paper is to capture the forecast-based decision-making landscape in which EA Duty Officers operate to understand how they perceive the potential impacts of this transition on their decision-making activities. To this end, a series of interviews were carried out in the summer 2018 with EA MFDOs and FWDOs. We hypothesise that the EA Duty Officers' decision-making is still very binary and that many elements of their decision-making process will have to change to make space for the transition to probabilistic fluvial flood forecasts.

After describing the interview and analysis methods (Sect. 2), this paper relates how EA MFDOs and FWDOs make forecast-based decisions in the current EA practice – with a focus on fluvial flooding and decision-making for up to the next 5 days, as these are the forecasts that will change (Sect. 3). Finally, based on the interview results and further literature findings, we discuss the Duty Officers' perceived opportunities and challenges associate with this transition and make a series of recommendations to support a successful transition to operational probabilistic fluvial flood forecasting at the EA.

## 2 Methods

### 2.1 Participants

The EA operates over 14 different areas (i.e. broadly based on catchment delineations) with 7 forecasting centres (hereafter referred to as 'centres'; see Fig. 1). Within these centres, it has several 'Monitoring and Forecasting Duty Officer' (MFDO) and 'Flood Warning Duty Officer' (FWDO) roles, fulfilled by a number of different people. These are voluntary roles, added to the staff's day-to-day job, for which they follow relevant training. MFDOs receive, process and communicate forecast information to FWDOs, who are responsible for interpreting the information and working out the potential impacts on the ground. The Duty Officers' schedules are predetermined by a rota, and Duty Officers are on call for a period of one week at a time. During times of increased flood risk, when more forecasting or warning activities are required, additional rostering takes place. Duty Officers receive a range of forecasts (nowcasting* products to monthly outlooks*) and are aware of potential situations from a month out. Five days ahead is when the activity really starts to build and is the focus of these interviews.

A total of six EA MFDOs and FWDOs from three different EA centres (one pair per centre) were interviewed to capture a range of perspectives in relation to the topic at heart, following best practice (Sivle et al., 2014; participant information sheet provided as supplementary material). Forecasting and decision-making varies between EA centres due to different management approaches and different types of geography and catchment response*. To protect anonymity, the three centres where interviews were carried out are shown in terms of the wider area they are responsible for: 1) the Yorkshire area (YOR) in the North (area 3), 2) the Thames area (THM) in the South East (area 11), and 3) the Solent and South Downs area (SSD) in the South East (area 14) (Fig. 1).

MFDOs and FWDOs were interviewed in pairs as they are used to working together and the information they use sits between these two roles. The thought was that by talking to the MFDOs alone we would lose the element of "and so what?", while talking to the FWDOs alone the forecasting expertise would be lost. All MFDOs and FWDOs interviewed had several years of experience and so were able to describe the current practice and express personal expectations of how it might change with probabilistic forecasting.

Participants were selected by EA study co-developer I1 to meet the above criteria. For the purpose of anonymity, the interviewees will thereafter be reported using codes. The three MFDOs interviewed will be referred to as MFDO1, MFDO2 and MFDO3, and the three FWDOs interviewed as FWDO1, FWDO2 and FWDO3 (interviewed pairs are represented by the same number). As well as those from the MFDOs and FWDOs, quotes from two EA study co-developers are reported in this paper, I1 and I2, who helped the interviewer (Louise Arnal) by providing some context about the EA's organisational landscape, forecasting systems and MFDO and FWDO roles prior to the three interviews.

**2.2 Interviews**

This paper is based on content gathered through interviews at the EA. Interviews can be an effective method to capture an institution's complex cultural landscape (Schoenberger, 1991; Pagano et al., 2004). They can provide interviewers with an understanding of the world (in this case the institution world) from the perspective of the informants, shedding light on their unique perceptions and information only known to them (Sivle et al., 2014).

Qualitative semi-structured interviews were carried out. These types of interviews are often used to understand interviewees' perspectives and allow the exploration of a research question that does not necessitate quantifying information and creating generalisations from the interview transcripts. The strength of such studies (compared to other survey methods) is that they are more sensitive to historical and institutional complexity and can capture the influence of local context (Schoenberger, 1991; Pagano et al., 2004). Moreover, they are flexible, allowing the interviewer to remodel questions throughout an interview and from one interview to the next, to follow up on new information discovered (Sivle et al., 2014).

A set of open-ended questions were prepared in advance to guide the discussion and allow for comparability across all three interviews. To prompt discussion, all three MFDO and FWDO pairs were asked the same opening question: "Could you please walk me through what you would do ahead of a potential flood event?" The following questions were also prepared in advance, but their order was changed, or they were skipped depending on whether the interviewees had already answered them:

- "Could you tell me about the uncertainties in the information you said you used in this context?"
- "How do you deal with these uncertainties?"
- "Could you tell me about how you communicate these uncertainties to each other?"
- "How would your job be influenced by a transition to probabilistic forecasts?"

Each interview lasted between 30 minutes and 1 hour 30 minutes. All interviews were conducted and digitally recorded by the first author (Louise Arnal) in meeting rooms at the corresponding EA centres.

## 2.3 Data analysis

All interviews were transcribed verbatim and transcripts were analysed qualitatively in order 1) to define the current practice (i.e. EA Duty Officers' roles and the information and systems they use) and 2) to explore, together with the EA Duty Officers, how they think this transition might affect their roles and the current practice. These two points provide the structure for this paper; the results are communicated in Sect. 3 and 4 respectively.

Although interpretations might have been communicated by several interviewees, no frequencies are provided as quantitative generalisations cannot be inferred from this small and purposive sample. Following best practice, the results contain a mix of interviewees' perspectives, supported by quotes (numbered for cross-reference), and further interpretation of the interview transcripts by the authors, identifiable throughout the text (Rowley, 2012; Davies et al., 2014).

## 3 The current EA forecast-led decision-making practice

### 3.1 The EA's institutional landscape

The schematic on Fig. 2 displays the EA's institutional landscape, with a particular focus on the flood incident management information flow to and from MFDOs and FWDOs. To help manage flood risk, the EA receive hydro-meteorological forecasts produced by the Flood Forecasting Centre (FFC) daily (more or less frequently depending on the forecasting product* – see Sect. 3.2.1). The FFC is a partnership between the EA and the UK Met Office which combines the hydrological and meteorological expertise from both institutes to produce hydro-meteorological forecasting products for all natural forms of flooding (including river, surface water, coastal and groundwater flooding; note that in this paper the focus is on river flooding). Within the EA, the FFC products are combined with the flood forecasting expertise of the Flood Forecasting team to then follow two separate routes. The flood forecast information is relayed higher up the chain to the Strategic Support and the National Response teams via the National Flood Forecasting Duty Manager to support national response. To support local response, the flood forecast information is downscaled to local flood outlooks by the MFDOs and passed on to the Area Response team, coordinated by Area Duty Managers and Area Base Controllers who are responsible for an area's incident preparation and response. As part of this team, the FWDOs and the Flood Incident Duty Officers then combine national information and area impact assessments to coordinate flood warning and operational decision-making on the ground, respectively. Pre-defined lead times are assigned to each specific planning and response activities; e.g. flood warnings are sent with a 2-hour minimum lead time, although different lead times have recently been introduced to account for flood event type and catchment characteristics* (i.e. flash flooding vs a slow responding catchment). FWDOs are also part of a Flood Advisory Service, an integrated service provided by the EA and the UK Met Office, via which the flood forecast is communicated with partners (e.g. emergency responders) to help them make informed decisions about their flood response. This service is delivered through emails, teleconferences and face to face meetings.

MFDOs and FWDOs are decision-makers* at the heart of the flood forecasting to local flood decision-making process, which relies vastly on the interaction between their two roles. Hereafter, we use the term "decision-makers" to refer to the MFDOs and FWDOs, unless stated otherwise.

**3.2 From national hydro-meteorological forecasting to local flood warning decision-making**

The sections below (Sect. 3.2.1 to 3.2.4) describe the flow of information between MFDOs and FWDOs in an incident response context (following the numbers on Fig. 2), ahead of a flood event (i.e. with 2 hours to 5 days of lead time, as this is the timescale likely to be affected by this transition). The content for these sections is based on the interviewees' responses to the question: "Could you please walk me through what you would do ahead of a potential flood event?" It is worth noting that all interviewed pairs suggested the MFDO answer that question before the FWDO, indicating that the forecasting and decision-making process starts with the MFDO.

> *"My role's an MFDO so generally if there's a flood event coming I should know before the FWDO, in theory" [MFDO2]*    (Q1)

**3.2.1 The FFC national hydro-meteorological forecasts**

The FFC generate three types of products of relevance for river flooding (i.e. coastal and high tides reports are also produced but not discussed in this paper; Defra, 2014). These are produced with the help of and communicated with the EA.

- **Flood outlook products:** annual, seasonal and monthly assessments of flood risk produced up to every two weeks (please note that these are not the focus of this paper; see paragraph below);
- **Flood Guidance Statement (FGS)\*:** a five-day forecast of flood risk for all sources of flooding, for England and Wales, at a county scale (i.e. area sub-divisions) and issued daily (with additional issues when significant or severe impacts forecasts; see Appendix B, Fig. (a) for a past example);
- **Hydro-Meteorological Services\*:** detailed products communicating flood forecast data, comprising , among others, a Hydro-meteorological Guidance (i.e. a summary of the hydro-meteorological situation for the next five days and issued once daily; Appendix B, Fig. (b)), Forecast Meteorological Data (i.e. rainfall summary for the next five days based on the 'Best Estimate' and issued twice daily; Appendix B, Fig. (c)), a Rainfall Scenario Map (i.e. 'Reasonable Worst Case' scenarios of rainfall amounts for areas across England and Wales; Appendix B, Fig. (d)), and Heavy Rainfall Alert for the next hours to five days (i.e. produced manually for specific rainfall events and communicating the rainfall amounts and confidence; Appendix B, Fig. (e) and (f)).

Since 2007 (this vaguely corresponds with the summer 2007 UK floods), the lead time for which forecasts are shown and on which MFDOs and FWDOs can take action has increased from a few days to a few months ahead (i.e. based on the FFC outlook products). However, the outlook products are currently mainly used as supporting information, and the EA relies on the shorter-range forecasts (i.e. FFC five-day products) for their flood warning decision-making activities. This is consistent with findings from Neumann et al. (2018).

> *"So even from a month out now we're starting to become aware of potential situations […], but […] because […] most of our products […] are […] based on that five-day forecast […] that's when the activity really starts to build" [MFDO1]*    (Q2)

**3.2.2 Local flood forecasting by the MFDOs**

The MFDOs' role is to process the FFC forecasts (see Sect. 3.2.1) before communicating the local flood forecast to the FWDOs.

> *"Ramping up to a flood event, the MFDO gathers that information, processes it and filters it, and passes that along to the area staff [FWDO]." [MFDO2]* (Q3)

Based on the suite of FFC national and county scale flood risk information, the MFDOs decide whether they should run the locally tailored hydrological forecasting model, which sits in a separate system called the National Flood Forecasting System (NFFS; Appendix B, Fig. (g)), to produce catchment/local scale flood forecasts. This decision can for example be triggered by the colours shown on the FGS, which communicates flood risk as a combination of likelihood and impact (i.e. high flood risk values on the FGS are likely to prompt the MFDOs to run the hydrological model). The NFFS allows users to explore observed data (i.e. river levels and rainfall) and run hydrological and hydraulic models*. These models, forced with the FFC's deterministic weather forecast, provide a single trace of future (i.e. for the next five days) river level at specific locations. This initial forecast scenario is usually referred to as the 'Best Estimate' scenario. According to the FGS user guide, the 'Best Estimate' scenario shows what 'should' happen, it is "a forecaster's assessment of the middle range of rainfall, river or groundwater levels or coastal conditions and impacts that may occur" (FFC, 2017).

What 'could' happen (i.e. referred to as the 'Reasonable Worst Case' scenario) may not always be run by the MFDOs. This decision is usually based on the hydro-meteorological conditions and on the MFDOs' expert judgment.

> *"If there's uncertainty in the forecast like if there's showers [...] especially when they're thundery and they can give you really high totals in a very short space of time that's when you start to run 'What If' scenarios" [MFDO1]* (Q4)

'What If' scenarios refer to the additional local river level forecast run by the MFDOs. This is usually done by manually modifying the FFC's deterministic weather forecast, using predefined factors applied over an entire catchment (e.g. a 200% increase in catchment rainfall totals over the next 6 hours). The MFDOs choose which 'What If' scenario to run based on the FFC Hydro-meteorological Guidance, the Rainfall Scenario Map and their own expert judgment.

> *"[The FFC] might give us a number of different scenarios and we tend to pick the worst one and then see what that does" [MFDO1]* (Q5)

Running this 'modified weather forecast' through the hydrological/hydraulic models, the MFDOs obtain a supplementary river level forecast scenario to the 'Best Estimate', called the 'Reasonable Worst Case' scenario (Appendix B, Fig. (g)). According to the FGS user guide, the 'Reasonable Worst Case' scenario shows what 'could' happen, it is "a forecaster's assessment of the upper range of rainfall, river or groundwater levels or coastal conditions and impacts that may occur" (FFC, 2017). The MFDOs estimate the likelihood of both scenarios (i.e. the 'Best Estimate' and the 'Reasonable Worst Case') based on the 'What If' scenario they have run and further expert judgment.

A critical part of the MFDOs' role is to filter the forecast information to make a coherent story (e.g. there may sometimes be differences between the national and local scale pictures) and put the information into context for the FWDOs. They do this using additional tools and information available to them and by applying expert judgement based on their knowledge of model performance* and catchment response*.

> *"Whilst we are very data reliant on the information coming through, there's also that experience that you know that certain watercourses are very slow responding and [...] no matter how much money we spend on your forecast, it's always not very good, you always delay it by a day and drop the peak by a bit. [...] Data is very important but that local experience is as important if not more so in certain circumstances" [MFDO2]* (Q6)

Additional tools and information available to the MFDOs for example include river level correlations* (i.e. they are calculated using set tables, based on a linear regression between peak levels upstream and downstream of a station). These complement the river level forecast and aid MFDOs in the decision-making process. However, discrepancies between the forecasts and correlations are possible and can call into question the forecast accuracy*.

> *"If the model says you're going to get flooding, the correlation says we're going to get flooding, we've* (Q7)
> *had more rainfall than any previous event, you know that that decision's [...] a clear one. If the model* 
> *says flooding, the correlation says no you're fine, and we've had somewhere in the middle in terms of* 
> *rainfall, that's when it gets difficult, because those borderline calls are really tricky to make"* [I2]

The MFDOs' knowledge of the hydraulic/hydrological model performances for various types of events and catchments is also key in interpreting the river level forecast. This can be based on experience, performance measures*, the FFC meteorological products' attached confidence, target lead times (i.e. the theoretical maximum lead time there is to send out a flood warning for a catchment before it floods, pre-calculated for each catchment based on its size, the gauge location and flood risk in that catchment) and local feedback from real time river gauges*.

### 3.2.3 Interaction between MFDOs and FWDOs

Weighing the various sources of information available to them, the MFDOs generally flag a situation to the FWDOs once they are confident* about the signal shown by the river level forecast. The exact content of the communication depends on each MFDO-FWDO pair, but usually contains information about the scale of the event and their confidence in the forecast (see Appendix B, Fig. (h) for an example).

> *"Which scenario is going through which threshold [and] how likely that is to happen"* [MFDO1]. (Q8)
> *"Approximate [...] scale of the event [...] are we talking just a bit of water out of bank? Or is it* 
> *Armageddon?"* [MFDO2]

> *"I look at the river level forecasts and then what I want to know from the MFDO is, does this account for* (Q9)
> *the rain we've had? So, do you think this is likely to change? Is the forecast I'm seeing on my screen a* 
> *good river level forecast? Or do we think it's not picked something up properly?"* [FWDO2]

The conversation can sometimes be bilateral, and the MFDOs might also ask questions to the FWDOs about local conditions.

> *"Can they [the FWDO] provide information [...] in terms of local sensitivity [...] and are works going on* (Q10)
> *in that catchment? Is there a gauge out of play?"* [MFDO2]

The communication between MFDOs and FWDOs varies across people and EA centres. Factors that might influence communication – in terms of its trigger, frequency and content – include the type of event, Duty Officers' geographical proximities (i.e. communication in person, by email or phone), a centre's practice, the Duty Officers' personality, day-to-day job and level of experience. Some FWDOs are more proactive than others in obtaining the information needed to make a decision; while some might wait to be contacted by the MFDOs with a processed forecast, others monitor the situation daily. In some cases, the FWDO might contact the MFDO first to get more details about an area of concern to them.

> *"The FWDO shouldn't even really be thinking about anything until they've had a phone call from the* (Q11)
> *MFDO [...]. Some FWDOs do go a bit more proactive than that, I think particularly the ones with the* 
> *forecasting backgrounds almost can't help themselves looking into it. And it depends on personality as*

*well, some people hate the idea of being surprised by anything. But it does also depend on the MFDO."*
*[FWDO2]*

*"[...] and [...] then it's [...] liaising with regional forecasting [the MFDOs] so they can give us any more* (Q12)
*detail or certainty or if we're concerned about an area they can watch it a bit more for us [the FWDOs]"*
*[FWDO3]*

The Duty Officers' level of experience can influence the content and interpretation of their conversation. Duty Officers who have been working together for a longer time will have more ease to interpret and gauge the confidence from each other's language, while working with new Duty Officers can sometimes lead to misinterpretations.

*"Knowing each other is really important because if I know it's [MFDO2] on duty [they've] probably put* (Q13)
*that interpretation on already. If I get someone who's reading off the screen, I put the interpretation on*
*and if we misjudge that and we both put it on we could end up getting it too low" [FWDO2]*

As can be seen with quote Q8 (i.e. the title of this paper), the forecast communication process is currently binary to an extent. However, as hinted by quote Q9, confidence and uncertainty in the information appear to be communicated between the MFDOs and FWDOs, usually using the two flood forecast scenarios. Understanding how the uncertainty will impact the FWDOs' decision-making is key.

*"I don't think we can withhold uncertainty. One, the key role for MFDO is providing the forecast. So it's* (Q14)
*getting the forecast as accurate as you can and then communicating it in the clearest way possible. So*
*that's often about interpreting the uncertainty and communicating it. So we often use the 'Reasonable*
*Worst Case' and the 'Best Estimate' to do that" [MFDO1]*

*"Uncertainty is present in everything that we do and every bit of communication, [...] I don't think I've* (Q15)
*ever been able to say something with 100% confidence, ever." [MFDO2]*

*"Uncertainty from the forecasting point of view is always prevalent but understanding how it will impact* (Q16)
*the [...] area's reaction is kind of the key thing" [MFDO2]*

### 3.2.4 Local flood warning decision-making by the FWDOs

The FWDOs' role is to combine the MFDOs' processed local flood forecast with local information to decide whether to issue a flood alert or warning.

*"The role of the FWDO is to make sense of all that forecasting information and try and work out potentially* (Q17)
*what the impacts could be of that on the ground and then make decisions as to whether or not [they] issue*
*flood alerts, flood warnings or severe flood warnings." [FWDO1]*

The information available to FWDOs includes:

- **The local flood forecast and its interpretation:** produced by the MFDOs (see Sect. 3.2.2 and 3.2.3);
- **Factors within the catchment that could influence river levels:** e.g. blockage from a fallen down tree. This is ad-hoc information and comes from a variety of sources, including: information gathered by community contacts (e.g. flood wardens* and flood action groups*), by EA staff and Duty Officers, hydrometric data/CCTV images, details of consented works (i.e. work going on in a channel);

- **The situation on nowcasting meteorological products:** e.g. rainfall radar;

- **Information about the catchment(s) that might be affected:** contained in the 'Flood Intelligence Files', available for every gauge of the NFFS. They compile catchment information such as: highest events on record, what rainfall led to them, what the catchment state was at the time and any known impacts;

- **Information about communities that might be affected:** e.g. have they been affected by many floods in the past.

The FWDOs combine and assess these various sources of information (i.e. in terms of their accuracy and uncertainty; according to *FWDO2*, a critical part of the FWDOs' role is the *"interpretation of the uncertainties"* into their impacts on the ground), together with their expert knowledge about catchment response, to make a judgment call on whether to issue a flood alert/warning. In some areas however, the MFDOs will tell the FWDOs when they need to issue a warning.

According to EA internal guidelines on using the two flood scenarios in practice, the 'Best Estimate' should be used as a basis to issue flood alerts and warnings, and the 'Reasonable Worst Case' should be used for incident planning activities (e.g. resources needed for response). However, both scenarios are currently used for incident planning and communication with responders and communities, while flood alerts and warnings are mostly issued based on nowcasting products. This discrepancy could be due to the challenges associated with forecast accuracy and lead time, specifically for rapid-response

catchments*. EA guidelines however encourage the use of the two scenarios for planning and flood warning activities whenever possible, in combination with expert judgement.

> *"The scenarios are planning scenarios and at some point [...] we move into operational now type* (Q18)
> *forecasting. So normally we'd issue a flood warning with anywhere between 30 minutes to [...] six hours*
> *lead time, whereas these scenarios are generally two to five days ahead. So you wouldn't normally [...]*
> *come up with a simple statement that will issue flood warnings based on the best estimate [...] and at some*
> *point we transition into something that's more now that we use for operational decision making" [I1]*

Warning procedures can vary across Duty Officers and EA centres. FWDOs' risk appetite (i.e. issuing too many or not enough warnings - risk-averse vs risk-hungry; which may be triggered by past events) can influence the decisions taken.

> *"Since the Boxing Day floods I think the next level of flooding after that there was some discrepancies* (Q19)
> *amongst the area responses [...] they were a bit [...] jumpy [...] to not be caught out again which is*
> *understandable" [MFDO2]*

Some areas and EA centres might be more forecast-led while others are more reliant on a nowcasting type approach.

Discrepancies amongst responses are partially due to historical differences across areas and EA centres, which could in turn be a consequence of differences in catchment characteristics (e.g. catchment size, rainfall-runoff response time, land use) and differences in typical response times (i.e. time for emergency responders to respond to a flood warning in a given area; also partly dependent on catchment characteristics).

> *"There are definite differences between areas and [...] between individual staff, so [town X] are far more* (Q20)
> *likely to issue flood alerts [...] purely on rainfall than [town Y] is, [town Y] will generally wait for a river*
> *level to rise and that develops I suppose out of slight historical differences and personalities involved"*
> *[FWDO2]*

> *"Some other areas will issue messages based on forecast whereas, we were always told to base it on* (Q21)
> *what's happening, so we kind of wait to see if the rain comes in and then if anything happens issue. And*
> *we get marked on messages that we send out, so one of the things is the timeliness and if you've issued*

*one, did it actually flood afterwards? So if you're obviously issuing on a forecast, then you're probably going to get scored low because it doesn't always happen, so it's difficult" [FWDO3]*

Please note that Duty Officers do not formerly get scored based on decisions taken, quote Q21 is a figure of speech. The flood warnings issued are nevertheless captured in the Flood Warning Validation Data Base, where a score is given to each warning based on whether flooding occurred, was missed, or the warning was sent out too late, etc. This database however does not capture catchment conditions or forecasts produced at the time the warning was issued (Susan Manson, personal communication).

There are additionally exceptions to the warning procedures for certain types of events and depending on the time of day a flood is expected to occur. For given types of events, such as convective rainfall events*, for which the Duty Officers know models are still limited, they might decide to issue a warning based on the 'Reasonable Worst Case', although it is *"technically against procedure" [MFDO2]*. FWDO3 also mentions the possibility of issuing flood alerts based on the forecast (see quote Q21) when the impact is expected to occur overnight or if the forecast displays *"rarely high confidence"* of rainfall and *"if it's a more prolonged event"* and *"the catchment's already wet"*.

Other factors that may cause the FWDOs to deviate from standard warning procedures may be political. There is for example usually a political element to the response immediately following a very major flood event, as the EA puts a greater focus on demonstrating to communities and the government that they are being proactive in warning, informing, etc. There is also the need for the EA to align its message with actions of lead local flood authorities and responders and to think about public response.

> *"There are lots of external pressures as well, particularly as FWDO you can come under pressure from all different types of sources to make decisions and perhaps not based on the evidence that you've got for political reasons, [...] reputational reasons, organisation, in terms of being seen to be active, seen to [...] act early" [FWDO1]*  (Q22)

> *"It's managing expectations internally in terms of operational response and how this is going to potentially play out which [...] can still be quite hard to do but it's even harder to do it externally with [the] mood of the public or even some of our professional partners, so local authorities are also obviously geared up to respond to flooding" [FWDO1]*  (Q23)

The EA's principle, 'think big, act early, be visible', might also influence the Duty Officers' decision-making (EA, 2018a). In what ways does the EA's statutory warning responsibilities and principle influence decision-making? Does 'act early' put the forecast in first place while 'think big' and 'be visible' move it to a secondary position?

> *"Our mantra to incident response is think big, act early so sometimes [...] there is a danger that you're over responding. Somewhere you're issuing alerts and warnings when actually the risk is low. So I think the role of the FWDO is to assimilate all that information, forecasting information and using it to help inform the instant response but also manage expectations" [FWDO1]*  (Q24)

Messages to the public are worded with care to communicate the appropriate level of risk and prompt appropriate response and also contain some information about confidence and uncertainty (see Appendix B, Fig. (i) and (j) for past examples of an EA flood warnings and alerts map and an EA fluvial flood alert message, both produced for and available to the public). As stated by internal EA guidelines, the language used should change according to each scenario. This can be seen as a step towards communicating probabilities.

*"If messages around a 'Reasonable Worst Case' use, could or [...] is possible; if it's a 'Best Estimate'* (Q25)
*use, we expect, it's probable" [I1]*

*"To help them [Duty Officers] get used to the language and the way they're working around scenarios* (Q26)
*and probabilistic forecasting" [I1]*

Public messages are usually free-text messages and will therefore vary across FWDOs.

*"The message starts off with this flood warning has been issued for this place then it runs on after a while* (Q27)
*into detail which is where you can communicate those shades of grey" [FWDO2]*

To conclude this section on the current EA practice, it is evident that, while forecasting supports incident response by providing a critical piece of information, Duty Officers have to make trade-offs, taking different sources of information into consideration for their decision-making process.

*"Forecasting's really important. It is, it should be really central to what we do [...] but actually it's a* (Q28)
*small cog in the middle of a much bigger wheel." [I1]*

*"We always implore people to try and look at different sources of information" [I2]* (Q29)

Additional sources of information and factors include river level correlations, model performance, local knowledge, personal experience, internal procedures and politics (see Fig. 3). However, the forecast helps determine the timing of warning and response activities. Because the forecast is a piece within a much bigger system, will the transition to probabilistic forecasting have very minor impacts on the Duty Officers? Or on the contrary, could it unsettle this very complex machine?

## 4 Duty Officers' perceptions on what the future practice might look like

The transition to probabilistic forecasts is a significant evolution, which will undeniably bring some changes at the EA, and generates mixed feelings amongst the Duty Officers.

*"Whether it creates as many problems as it solves, maybe" [I2]* (Q30)

*"Probabilistic forecasting is kind of a fresh start for everyone" [FWDO2]* (Q31)

This section presents the interviewed Duty Officers' perspectives (as quotes) on what the future practice might look like for EA Duty Officers. In light of these findings and relevant literature findings, we make a list of recommendations to support the uptake of probabilistic forecasts at the EA. These recommendations concern actions we think the EA should take with high priority. The service, role owners and those responsible for ensuring a quality service delivery should ensure that these recommendations are pursued, alongside technical work around the transition. Please note that these recommendations are not ranked in priority order for the EA, as some of these will be quicker and easier to implement and to demonstrate progress on.

### 4.1 The FFC national hydro-meteorological forecasts

At the time of the interviews, very little was known to Duty Officers about the new probabilistic forecasting system (which research and implementation project started in November 2008; Sene et al., 2009; 2010). However, at least one interviewed MFDO was involved in the future system's technical implementation.

While the new system's design was not formerly known by all yet, some interviewees think that the probabilistic forecast could help materialise the uncertainty otherwise sometimes hidden with the two scenarios.

*"I think in a good way [...] it will [...] reveal the uncertainty that's hidden by apparent simplicity" [I1]*     (Q32)

Many interviewees however seem concerned that the probabilistic forecast could add another layer of uncertainty to their already uncertain decision-making process (see Sect. 3.2).

*"Uncertainties are very tricky to deal with, whether probabilistic forecasting and a switch to that is going to help?" [MFDO2]*     (Q33)

*"That would be my concern that it's even more information and more uncertainty and it's kind of like, well what do you do with this information? And which bit do you communicate to who?" [FWDO3]*     (Q34)

From these interviews, it is apparent that in the current practice (see Sect. 3.2) Duty Officers see uncertainty as an inherent component of their decision-making process (see quotes Q14-16) and appreciate that forecasts convey uncertain information, not unlike other types of information they use, which they currently communicate using the two flood scenarios. Decision-makers "view uncertainty as an unavoidable factor [...,] all information about the future is uncertain [and] they must make decisions under uncertainty every day" (Morss et al. 2005). This is in line with the positive perception captured in quote Q32 about probabilistic forecasts revealing otherwise hidden uncertainty. In fact, numerous studies have shown that decision-makers want to see that uncertainty, which they do not necessarily perceive as a barrier to use (Morss et al., 2005; McCarty et al., 2007; Bruen et al., 2010; Neumann et al., 2018). Ramos et al. (2013) have additionally showed that providing uncertainty attached to a forecast leads to more optimal and consistent decisions across decision-makers; when decision-makers "are not provided with estimates of forecast uncertainty they attempt to take uncertainty into account on their own" (as hinted by quote Q13 about the current practice), which may lead to important errors and/or risk-averse decisions (Joslyn and Savelli, 2010; Joslyn et al., 2011; Ramos et al., 2013; Michaels, 2014).

However, while they appreciate that the information conveyed by forecasts is uncertain, many Duty Officers expressed worries about the consequences this 'visible uncertainty' may have on their decision-making process as they perceive this transition as an increase in information (see quotes Q33 and Q34). Mu et al. (2018) looked at decisions taken by participants based on the UK Met Office weather risk matrix, with varying information content and format, and concluded that "while increasing the information with content of warnings is usually beneficial and increases the trust in the warning system, it must be done with caution since better decisions (judged by higher profits) are not always made with an increase of information." This was also put forward by Michaels (2014).

These trade-offs highlight the need for a careful design of the probabilistic forecasting system. Indeed, a great amount of research has explored the impacts of graphical representation of uncertainty in hazard forecasts on decision-making, and showed that the design and communication of uncertainty information can impact the nature of actions taken and should be tackled with care (Bruen et al., 2010; Joslyn and Savelli, 2010; Stephens et al., 2012; Pappenberger et al., 2013; Sivle et al., 2014; Mulder et al., 2017). In this context, opening a dialogue between forecasters, developers and end-users and allowing for all parties to be involved in the co-design of forecast products is vital (Morss et al., 2005; Smith et al., 2018; Fundel et al., 2019).

**Recommendation 1: While Duty Officers acknowledge the value of probabilistic for communicating otherwise hidden uncertainty, they are worried about the impact it may have on their decision-making process. We therefore recommend the expansion of existing EA communication structures to allow the co-design of the new probabilistic forecast products between FFC forecasters and Duty Officers. This will ensure that Duty Officers have a say in the new system's uncertainty visualisation and communication in FFC documents, and may help tackle some of their worries.**

The idea expressed in quote Q34, that probabilistic forecast means *"even more information and more uncertainty"* highlights a common misconception about probabilistic forecasting. Indeed, probabilistic forecasts do not add more uncertainty, but they offer a formal estimate of the uncertainty inherent to the information conveyed (e.g. river levels). This highlights forecast users' need for accurate information (as seen in the current practice; Sect. 3.2), which may seem to be at odds with probabilistic forecasting (McCarthy et al.; 2007). It further hints that there is still an important barrier between scientists/scientific notions

and forecast users, which the geoscience community should aim to tackle. As pointed out by Morss et al. (2005), "the way scientists referred to and discussed uncertainty sometimes confused practitioners".

> **Recommendation 2: The misconception that probabilistic forecasts add "more uncertainty" calls for the adequate training of EA Duty Officers on probabilistic forecasting. This training could be delivered by forecasters from the FFC or by institutes the EA is already working with (e.g. JBA Consulting, the University of**

> **Reading, the University of Leeds). In this context, using serious games may help deliver the right messages in an engaging setting (see the HEPEX[1] and the Red Cross Climate Centre[2] resources and the IMPREX online game[3] for inspiration).**

### 4.2 Local flood forecasting by the MFDOs

From these interviews, it is apparent that this transition is not perceived as particularly challenging by and for the MFDOs,

despite the changes it might incur.

> *"I think the MFDO role won't change, it will still be to communicate a forecast but the [...] wording of*   (Q35)
> *the forecast may change slightly"* [MFDO1]

> *"I think from our point of view it will just mean a bit more interpretation of forecasts and then [...] just a*   (Q36)
> *slightly different way of passing it on [...]. But I don't think it will change the process"* [MFDO3]

The interviewed MFDOs mentioned several potential opportunities they perceive this transition will bring. Some interviewees for example mentioned that the two scenarios, and the 'What If' scenarios used to produce them, were sometimes complicated to play with and required a lot of expert judgment, making them inconsistent nation-wide. A few MFDOs thought probabilistic forecasts might lead to more consistency across the EA centres.

> *"The new flood forecasting system is being developed at the moment so it's going to replace the NFFS.*   (Q37)
> *[The] benefits to that I suppose [...] are that if we can look to be more consistent across the country in*
> *even simple things like what displays look like [...] we're more interoperable if we need to"* [MFDO1]

According to an interviewee, probabilistic forecasts could help with new staff training by increasing their understanding of catchment response.

> *"I can see some benefits to it, especially when you've got less experienced staff [...], you're almost [...]*   (Q38)
> *showing them the breadth of what a catchment could do given a range of responses"* [MFDO2]

As showed by quote Q37, this transition is perceived as an opportunity for more consistency across EA centres. However, given the heterogeneity of the EA at a national level and the areas' diversity in terms of history and catchment response (as highlighted in the current practice Sect. 3), we do not expect probabilistic forecasts to be welcomed similarly in all the EA

centres. This is transparent in the variety of perceptions captured by quotes in this Sect. 4.

---

[1] hepex.irstea.fr/resources/hepex-games
[2] www.climatecentre.org/resources-games/games
[3] https://www.imprex.arctik.tech/

**Recommendation 3: Considering the existing differences in the current practice across EA centres, we recommend that the EA ensure a simultaneous transition in all its centres.**

As mentioned by Handmer and Proudley (2007), decision-making based on probabilistic forecasts can be challenging because of situation-specific factors, which vary greatly across events, EA centres and catchments as shown in the current practice
Sect. 3 and as suggested by quote Q38. In this context, Nobert et al. (2010) advocate training for ensemble prediction systems users to be locally tailored to the local experience of different audiences.

**Recommendation 4: To account for the heterogeneity of local conditions, existing dynamics and institutional practices across EA centres, we recommend that the EA carry out a locally tailored customised transition within each centre. Building on recommendation 1, we recommend the co-design of the probabilistic forecasting system with a panel of Duty Officers representative of all EA areas. Moreover, training and the Duty Officers' operating procedures should adequately reflect these local differences.**

### 4.3 Interaction between MFDOs and FWDOs

It is worth noting that none of the interviewees mentioned worries concerning potential impacts of this future transition on the communication and interaction between their two roles.

> *"Between us [Duty Officers], it's probably OK because we've got that understanding of the roles"* (Q39)
> *[FWDO3]*

This transition is perceived as an opportunity by MFDOs for more confidence and credibility when communicating with FWDOs.

> *"If you've got a huge spread then you know that there's a very wide range of impact potentially, but if* (Q40)
> *[...] everything's within a couple of centimetres of each other, it gives you a lot more confidence in saying,*
> *no I think we're going, we're not going to see a threshold crossing. So [...] it will help decision making I*
> *think" [MFDO3]*

In a study exploring the use of ensemble hydrological forecasts by decision-makers throughout Europe, Ramos et al. (2013) found that most decision-makers used the uncertainty information from the ensemble forecast to confirm the deterministic forecast. They observed that if the ensemble forecast showed a similar signal to the deterministic forecast, it made them more confident (as highlighted by quote Q40). If the two forecasts' signals differed it made them more "confused". This happens to some extent in the current EA practice, where discrepancies between the two flood scenarios and the river level correlations can call into question the forecast accuracy (see quote Q7). We can imagine that this could still happen in the future practice, if the probabilistic forecast shows a wide range of possible outcomes or a different signal to for example the 'Best Estimate' (if still in use).

However, the various information sources and tools that constitute the Duty Officers' current practice are vital in their decision-making process, as shown in Sect. 3. Indeed, as showed by Pidgeon and Fischhoff (2011), decision-makers may benefit "from different perspectives that help them clarify the implications of a decision on what they value"**.**

**Recommendation 5: To clarify how the probabilistic forecast should be used in combination with the tools and information sources Duty officers are already using in the current practice, we recommend updating the Duty Officers' operating procedures to contain specific guidelines about: the various sources of information officially available to Duty Officers for decision-making, how to interpret a probabilistic forecast, the forecast confidence at which given decisions and actions should be made and the language that should be used. The updated**

**guidelines should also describe a course of action to be followed by Duty Officers when the probabilistic forecast shows a contrasting signal to the deterministic forecast or other information sources.**

Duty Officers seem optimistic as to the impact this transition may have on their interaction. However, as mentioned by Michaels (2014), with "deterministic models it was easier to consider that a linear approach to forecast transmission was adequate", as was observed to some extent in the current practice as MFDOs generally flag a situation to the FWDOs once they are confident about the signal shown by the river level forecast (Sect. 3.2.3). We can imagine that the Duty Officers' interaction might become less linear with probabilistic forecast and may as a result require more comprehensive discussions

than it sometimes does in the current practice (Sect. 3.2.3).

**Recommendation 6: To lay foundations for adequate probabilistic forecast transmission from MFDOs to FWDOs, which we expect to be less linear than with deterministic forecasting, we recommend combined MFDOs-FWDOs training.**

### 4.4 Local flood warning decision-making by the FWDOs

This transition generated mixed perceptions with regards to the FWDOs role. Some interviewees believe that probabilistic forecasts will not solve the fundamental need of flood warning decision-making to be "binary" and see this transition as a potential cause for misunderstandings both internally and externally.

*"All the comms research we hear about generally says [...] the public message has to be as simple as possible, so that is working the opposite way to any proposal for probabilistic forecasting"* [FWDO2]  (Q41)

*"A lot of local authorities standing their staff up, putting them on standby for a weekend is quite a big budget thing [...]. So [...] if we say, it is going to flood, they can justify the spend on it [...]. If we pass it on as shades of grey, a lot of them, they'll appreciate the information but some of them would actually resent having the decision forced on them because they will struggle to then justify doing something or they'll be blamed, either way, blamed for spending money if it doesn't happen and blamed for not spending enough if it does happen."* [FWDO2]  (Q42)

*"You're still going to have this overriding issue with fast responding catchment where one scenario says we might need to issue a flood warning but 99 of them say no. Someone has to make a decision"* [MFDO1]  (Q43)

*"I think still for a lot of people the question they [...] want answered is am I going to flood?"* [I2]  (Q44)

Some interviewees also expressed the worry that probabilistic forecasts might push more of the interpretation further down the decision-making line and on to the FWDOs.

*"Having probabilistic forecasting just moves the burden of making a decision further down the tree"* [MFDO2]  (Q45)

*"I think my role is going to be the one where it has to stop and it can't be probabilistic because it [...] does come to a yes or no, issue it, don't issue it. So to some extent, probabilistic forecasting does feel like everyone else just pushing things down the line saying you make the decision, [...], we have to make the decision because we're the last ones on the line"* [FWDO2]  (Q46)

A few interviews however perceive this transition as an opportunity for early warning and long-term planning.

*"I think in an incident I'm happy that that's [...] a useful range of things to know, [...] you probably warn*      (Q47)
*for the lowest one and plan for the highest one and we can interpret between them" [FWDO2]*

*"We're talking about some of these decisions that have got a long lead time, we're going to move people*      (Q48)
*around the country, we're going to move equipment. It takes a long time to do that" [I1]*

It was also clear from the interviews that the transition needs to be gradual to give Duty Officers time to build confidence in the new system.

*"It is something to bear in mind with that if probabilistic forecasting put too much pressure and stress on*      (Q49)
*decision making on the people in these roles, the system probably would just collapse, people would walk*
*away" [FWDO2]*

The term "binary" decision which transpires through quotes Q41-44 and in the current practice quote Q8 may be deceiving. The FWDOs can make a range of different decisions (from incident planning to flood warning; see Sect. 3.2.4), and a decision is never "bad" nor "good", but should be the "best" that can be made at a point in time with the information available. In the face of a socio-political context that is demanding ever more precise information and with the rise of a post-factual society (i.e. culture in which public opinion depends on appeals to emotions rather than objective facts), the general trust in science might be a limiting factor to the uptake of probabilistic forecasts (Soares and Dessai, 2015; Golding et al., 2017; Knudsen and de Bolsée, 2019). Very often, the ability of an institution to pick up new information and methods is not only down to them, but could be influenced by the wider socio-political context and other key actors in the decision-making web (e.g. the government, local authorities, regulations and guidelines), additionally to the populations at risk and the way they respond to flood warnings (Dessai and Hulme, 2004; Morss et al., 2005; Parker et al., 2009). Michaels (2014) states that "conveying the uncertainties surrounding scientific knowledge and admitting the limitations of that knowledge helps gain and retain decision makers' and the public's trust". In the current practice, forecast confidence is already communicated to an extent to the public (see Sect. 3.2.4). This is a step towards probabilistic forecasting. But how big of a step is still needed to reach that full transition to probabilistic flood forecasts?

**Recommendation 7: To address a socio-political context demanding ever more precise information, we recommend that the EA, as the lead authority for flood warning in England, communicates (via engagement campaigns, videos, email newsletters, social media updates and webinars, etc.) with external players (i.e. emergency responders, the public and professional partners) ahead of this transition to ensure that they are aware that probabilistic forecasting will become operational practice and to communicate the benefits of probabilistic over deterministic forecasting. This includes rethinking the language that will be used in warning messages going out to public and, as part of the EA's strategic overview role, preparing external decision-makers on how this might change their practices.**

It is also important to note that "moving to probabilistic forecasting from deterministic forecasting may trigger an institutional shift in who is responsible for decision making under uncertainty" (Michaels, 2014). Because making a decision based on probabilistic information is more nuanced than using deterministic information, the outcome will determine who will be "blamed" and this ownership of the uncertainty judgment might have implications on the forecasters-users relationships (Michaels, 2014). With regards to the implementation of the European Commission's European Flood Awareness System (EFAS), sending flood alerts to European National flood forecasting agencies, it was noted that EFAS users were concerned about being held responsible for "wrong" EFAS alerts (Demeritt et al., 2010). This echoes our findings regarding some of the interviewed Duty Officers' fears regarding this transition potentially pushing more of the interpretation on to the FWDOs (see

quotes Q45 and Q46). This worry seems present in the current practice to an extent, as an interviewed FWDO expressed the difficulties of forecast-based decision-making and their concern of being "scored low" as a result (see quote Q21). Furthermore, those not involved in the probabilistic forecast production may not be comfortable with the responsibility of interpreting the uncertainty they convey (Faulkner et al., 2007; Michaels, 2014).

In this context, decision-making methods can provide a framework to support users in making a decision based on probabilistic forecasts, ensuring that decisions are made "objectively, with confidence and an understanding of uncertainty" (Dale et al., 2012). Examples of decision-making methods from the literature (see Duan et al. (2019) for more examples) include: 1) the series of decision-support methods developed by Dale and Wicks (2013), where the efforts associated with using each specific method is proportional to the costs and benefits of the decision at stake. The basic method associates a probability threshold to a specific flood incident management action based on expert judgement and local knowledge. The detailed method is based on a pre-defined water level-impact relationship used to determine, in real time, whether the average forecasted flood impact (if no action is taken) is greater than the flood incident management action cost (Dale et al. 2012); 2) a method for data-scarce locations, which links the latest forecast with an action based on the magnitude of past flood events and the decision-makers' willingness to "act in vain" (Coughlan de Perez et al., 2016); 3) the current EA method for decision-making based on ensemble surge forecasts, where extreme probabilities are communicated with responders who understand the low probability but need to mobilise out to 5 days ahead, and escalating or scaling down response closer to the event as the uncertainty narrows down (Gold and Connolly, 2018).

**Recommendation 8: Given the FWDOs' fears of having to interpret the probabilistic forecast and potentially being blamed for decisions made, we recommend the co-design of a tailored risk-based decision-making framework between the FFC and the EA Duty Officers. This will promote a co-ownership of the methods so that Duty Officers are more comfortable in interpreting and using the probabilistic forecast.**

"Institutional mandates understandably dictate what staff members emphasize" (Michaels, 2014). As shown in the EA current practice, decisions are sometimes led by internal procedures and politics (see Sect. 3.2.4). The cultural landscape in which decision-makers operate not only has an impact on the decision-making outcome, but may also have an impact on an institution's uptake of probabilistic flood forecasts in practice (Nobert et al., 2010; Ishikawa et al., 2011; McEwen et al., 2012; Demeritt et al., 2013; Michaels, 2014). Institutions like the EA have specific flood management priorities: seeking to avoid false alarms, or on the contrary, seeking to avoid missed flood events, and the minimum/maximum lead time at which they have to issue flood warnings. There is no doubt that probabilistic forecasts will offer a very different perspective on these factors. Bischiniotis et al. (2019) for example showed that the optimal lead time to trigger action depends on both the actions' operational implementation time and the probabilistic flood forecast quality. While the EA currently operates with pre-defined lead times for each specific planning and response activity (see Sect. 1), probabilistic forecasts could in theory provide earlier indications of potential upcoming floods, giving the EA more time to prepare ahead of a flood event. A few interviewed Duty Officers indeed perceive this transition as an opportunity for early warning and long-term planning (see quotes Q47 and Q48).

**Recommendation 9: To ensure an EA-wide successful transition in practice, we recommend that the EA adapt their wider flood management priorities. For example, the EA will have to be prepared to move towards lead times that reflect the probabilistic forecast predictability. To this end, tailored studies should be carried out during the system's co-design and implementation to identify new planning and warning lead times (reflecting the probabilistic forecast predictability – and therefore the event and catchment type -, actions' operational implementation time, and the EA's acceptable flood incident management action vs impact cost ratio). This should be done with ample time for testing by the EA Duty Officers.**

Morss et al. (2005) found that "although flood management practitioners might appreciate more certain hydro-meteorological information, scientific uncertainty is often swamped by other factors [e.g. community perception, time, money and resource constraints] and thus is not a high priority." When uncertainties are evident and decision stakes are high, as is the case for the uncertainty communicated by probabilistic forecasts for flood incident management, traditional decision-making pathways could become ineffective and soft values (dependent on culture, context and personal experience; see Sect. 3) might become more important than hard scientific facts (e.g. river level correlations, model performance and local knowledge to an extent; see Sect. 3) (Funtowicz and Ravetz, 1993). Given the complex decision-making landscape within which EA Duty Officers operate (see Sect. 3), this could translate to low probabilities of extreme events being ignored, which could have ultimately led to appropriate and earlier flood warnings.

Furthermore, facing constantly evolving soft values, some decision-makers may find familiarity with the scientific methods they use reassuring, reducing their personal willingness to adopt new scientific methods (Morss et al., 2005; Ishikawa et al., 2011). The interviewees' personal willingness was captured during these interviews, hinted by the range of quotes presented in this Sect. 4.

To ensure that users gradually adapt to a new system, there should be a reasonable period of overlap between two systems (Funtowicz and Ravetz, 1993). Additionally, as mentioned by Thielen et al. (2006; 2009) who documented the implementation of EFAS, the success of a new system should be measured via end-user feedback. used to update design and procedures throughout the system's operational implementation.

**Recommendation 10: To ensure a successful transition during which users can gradually adapt to the new system, we recommend a reasonable period of overlap between the two scenarios and the probabilistic forecasting system. During that time of overlap, end-user feedback should be collected from all key players and considered to update the new system's design and procedures.**

**4.5 Additional recommendations**

As mentioned in Sect. 4.1, while some interviewed Duty Officers knew about the transition to probabilistic forecasting and were involved in the technical design of the new forecasting system, a few interviewees had just learnt about the transition a few days prior to the interviews. This may help explain the diversity of perceptions presented in this Sect. 4 and some of the perceived challenges. Golding et al. (2017) identified the lack of user engagement as a great limiting factor of the uptake of climate information in practice. Ramos et al. (2010) additionally advocated the use of integrated platforms to allow a continuous exchange between scientists and decision-makers in real-time.

**Recommendation 11: In light of the Duty Officers' perceived challenges, building on recommendation 7, we recommend that the EA foster user engagement by putting in place a framework to communicate progress with all key players in the decision-making web (both internal and external) ahead of and during the EA's transition to probabilistic forecasts.**

Other factors that may help explain the diversity of perceptions reported by the quotes presented in this Sect. 4 are: the wording of questions that prompted them, personal resistance, and/or the interviewees' experience with the 2013/2014 transition from a single flood forecast to the two scenarios. These points were however not explored further during the interviewees and may be quite relevant for the prospects of the upcoming change from two scenarios to probabilistic forecasting. This merits additional investigation.

As stated in the introduction (Sect. 1), transitioning to operational hydrological probabilistic forecast is still a prevailing challenge in the field. Reaching out to the community of practice and institutes which have undergone this transition may help to gain insights and share best practice, as some elements might be transferrable (Nobert et al., 2010; Dale et al., 2012).

**Recommendation 12: To gain external insights on how to achieve a successful transition in practice, we recommend that the EA reach out to the community of practice in hydrological probabilistic forecasting, such as HEPEX (community of international experts in the field of probabilistic hydrological forecasting and decision-making) and connect with institutes which have already gone through such a transition, such as EFAS. This could for example be done through the establishment of an advisory group and organised workshops.**

Similarly, insights from this transition could be of great value to the EA for future transitions, to other institutes facing or yet to face a similar situation, and to the wider geoscience community to contribute to the advancement of the field of operational science (Pielke, 1997).

Post-event analyses can help improve a forecasting system, by identifying challenges in the current system and warning practices, and during the post-transition phase, by evaluating the impact of the introduction of a new system (Thielen et al., 2006). However, in the EA's current practice, while warnings sent out by the EA FWDOs are logged in a database, they are not currently used for post-event analysis (see Q21 and the associated text below it). This could have formed valuable insights for the current transition to probabilistic forecasts at the EA.

**Recommendation 13: To help future transitions at the EA and other institutes, we recommend that the EA document (in writing or through documentary-style interviews, etc.) and evaluate this transition (and the new forecasting system; via post-event analyses).**

## 5 Conclusions

The Environment Agency (EA) is currently transitioning from two flood scenarios to probabilistic fluvial flood forecasting for operational flood incident management in England. Probabilistic forecasts enable a better and earlier detection of potential future floods and their associated impacts, increasing the amount of time we have to prepare. However, there is currently a lack of clarity about how probabilistic forecasts should be used for flood incident management and how this transition will affect decision-makers' roles at the EA. To address this issue, interviews were carried out with EA 'Monitoring and Forecasting Duty Officers' (MFDOs) and 'Flood Warning Duty Officers' (FWDOs), two roles at the heart of the EA's flood warning decision-making chain. These interviews aimed to capture the Duty Officers' current decision-making process and their perceptions on how this transition to probabilistic forecasting might impact their decision-making activities. Based on these interviews and literature findings, thirteen recommendations were spelled out to support a successful transition for flood early warning in England. The interviews highlight the complex landscape in which EA duty officers operate and the breadth of factors that inform their decisions, additionally to the forecast. Within this landscape, the interviews revealed that, while EA Duty Officers already account for uncertainty and communicate their confidence in the forecast they currently use, their decision-making process is still very binary and the forecast transmission from MFDOs to FWDOs linear to an extent. This appears at odds with probabilistic forecasting and hints that several elements of the EA Duty Officers' forecast-based decision-making process will have to change. Key recommendations include (in no specific order):

- Enabling the co-design of the probabilistic forecasting system and a risk-based decision-making framework between forecasters and EA Duty Officers.
- Updating the Duty Officers' operating procedures with specific guidelines on how the probabilistic forecasts should be used in practice in combination with other tools they currently use.
- Communicating with internal and external players (i.e. emergency responders, the public and professional partners) that probabilistic forecasting will become operational practice and clarifying the benefits of probabilistic over deterministic forecasting.
- Adapting the EA wider flood management priorities (e.g. warning lead times).

**Data availability.** The participant information sheet, provided to the interviewees prior to the interviews, is included in the Supplement. The interview transcripts are not publicly available in order to protect anonymity. If readers require further information, this may be provided by contacting the corresponding author.

**Author contributions.** H.L.C., L.An. and S.M. posed the original question. L.An., S.M., T.N. and L.W. brought L.Ar. up to speed about the EA and their decision-making practices. T.N. identified the interviewees. L.Ar., H.L.C., T.N. and E.S. designed the interviews. L.Ar. carried out the interviews and analysed the interview transcripts. L.Ar., J.N. and H.L.C. wrote the paper. H.L.C., S.M., J.N. and T.N. commented on the manuscript. T.N. and S.M. provided L.Ar. with EA documents and forecast product examples.

**Competing interests.** The authors declare that they have no conflict of interest.

**Disclaimer.** The information and findings in this paper are based on interviewees with six EA Duty Officers. They should not be taken as representing the views or practice of the EA as a whole.

**Acknowledgements.** This work was funded by the EU Horizon 2020 IMPREX project (www.imprex.eu) (641811) and the joint Flood and Coastal Erosion Risk Management Research and Development Programme. We thank all the interviewees who dedicated some time to this work. We would also like to thank Stuart Hyslop at the EA for the background he provided in preparation for the interviews, and for his support in the organisation of the interviews. Lastly, we would like to acknowledge the invaluable help provided by Neil Ryan at the EA in sourcing the right materials for Appendix B.

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

© Crown copyright 2020

**Figure 1: Map showing the geographical areas of the EA's operations (green numbered areas), highlighting the three areas which the centres where interviews were carried out are responsible for (blue boxes) (source: EA). The works published in this journal are distributed under the Creative Commons Attribution 4.0 License. This licence does not affect the Crown copyright work, which is re-usable under the Open Government Licence (OGL). The Creative Commons Attribution 4.0 License and the OGL are interoperable and do not conflict with, reduce or limit each other.**

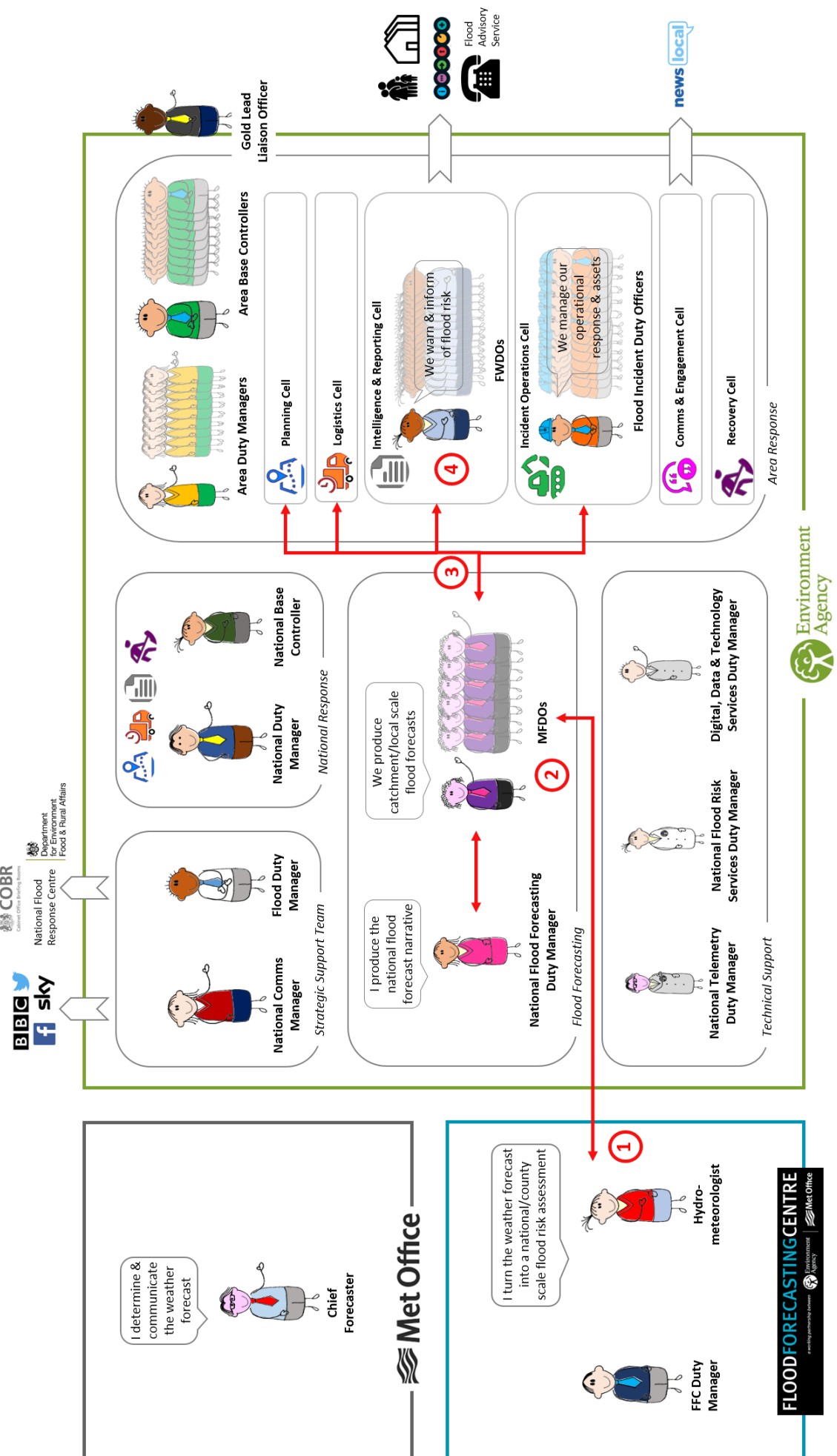

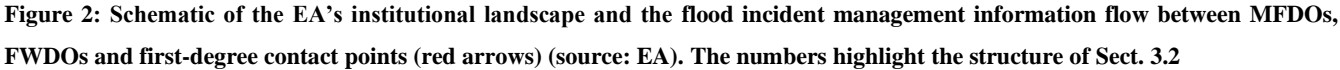

**Figure 2: Schematic of the EA's institutional landscape and the flood incident management information flow between MFDOs, FWDOs and first-degree contact points (red arrows) (source: EA). The numbers highlight the structure of Sect. 3.2**

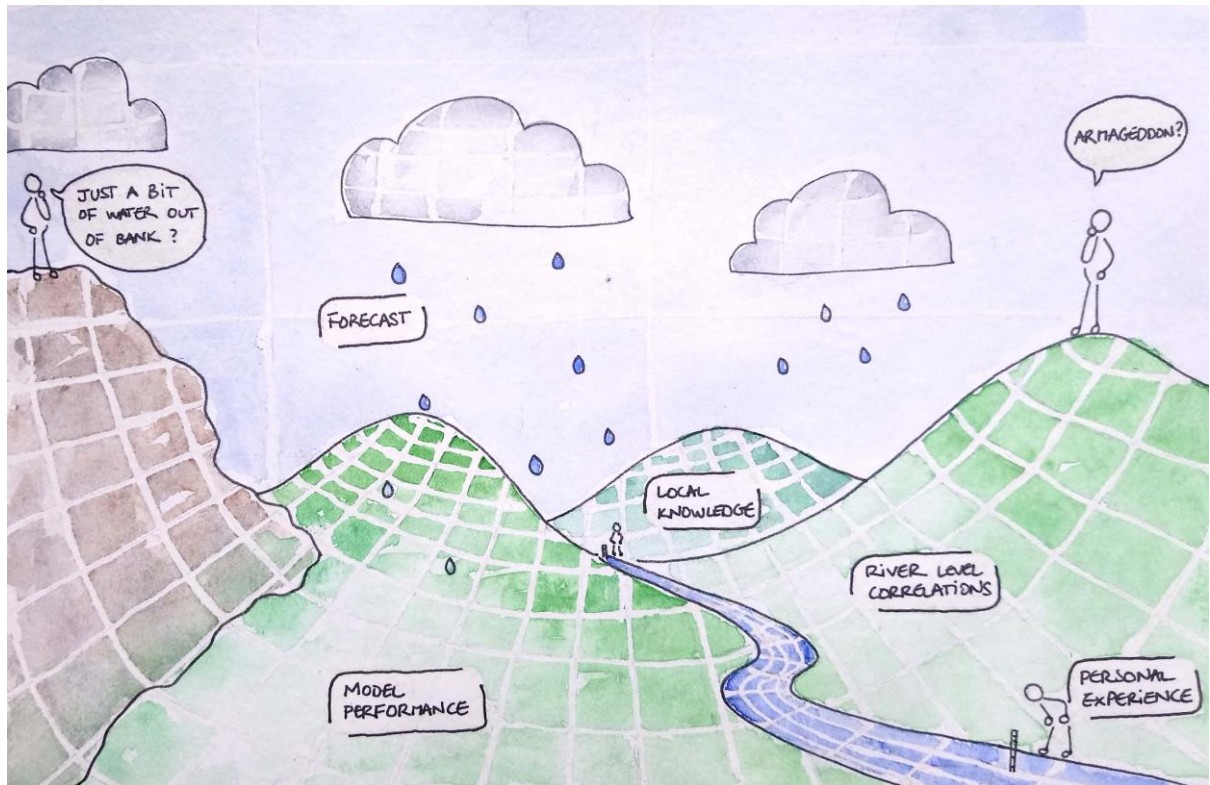

**Figure 3 Complex flood forecast interpretation landscape in which EA Duty Officers operate.**

**Appendix A. Glossary of terms.**

| | |
|---|---|
| **Best Estimate** | A forecaster's assessment of the most likely rainfall, river and groundwater levels, and coastal conditions, and their impacts. |
| **Catchment characteristics and response** | Catchment characteristics are the features that describe a river basin (i.e. the area of land drained by a river), such as its location, size, vegetation cover, soil type and topography. They partially define the catchment response, the catchment's reaction when subjected to a rainfall event (e.g. how fast the water level increases after a rainfall event). |
| **Confident** | A forecaster's expert judgement of how certain they are that the forecast is right, combining various sources of information (e.g. model performance information). Please note that in the literature, the term "confident" may also refer to the uncertainty range of a prediction, where a "confident" forecast is an ensemble forecast with a small uncertainty range. |
| **Convective rainfall events** | The sun heats the ground, warming up the air above it. This causes the air to rise. As the air rises it cools and condenses, forming water droplets that organise into clouds and lead to rainfall. Convective rainfall events can lead to thunderstorms. |
| **Decision-makers** | Persons whose professional role requires them to make important actionable decisions based on one or multiple sources of information. |
| **Department for Environment, Food and Rural Affairs (Defra)** | UK government department responsible for safeguarding the UK's natural environment and supported by 33 agencies and public bodies, including the Environment Agency (EA). www.gov.uk/government/organisations/department-for-environment-food-rural-affairs |
| **Deterministic forecast** | Refers to a forecast which gives a single possible outcome of the future rainfall, river and groundwater levels and coastal conditions. |
| **Ensemble** | Instead of running a single deterministic forecast, computer models can run a forecast several times, using slightly different inputs to account for uncertainties in the forecasting process. The complete set of forecasts is called an 'ensemble', and each individual forecast within it are 'ensemble members'. Each ensemble member represents a different possible scenario of future rainfall, river and groundwater levels and coastal conditions. Each scenario is equally likely to occur. |
| **Environment Agency (EA)** | An executive non-departmental public body sponsored by Defra. The EA has an operational responsibility to manage risks of flooding from rivers and the sea in England, by warning and informing the public and businesses about impending floods. www.gov.uk/government/organisations/environment-agency |
| **False alarms** | A warning given ahead of an event (e.g. flood) that does not ultimately occur. |
| **Flood action groups** | Cores of local people who act as representative voices for their wider community. They work alongside agencies and authorities and meet on a regular basis with the aim of reducing their community's flood risk and improving its resilience to flooding. |
| **Flood Forecasting Centre (FFC)** | A partnership between the Environment Agency and the UK Met Office. It provides a UK-wide 24/7 hydro-meteorological service to emergency responders to better prepare for flooding (river, surface water, tidal/coastal and groundwater). www.ffc-environment-agency.metoffice.gov.uk |
| **Flood Guidance Statement (FGS)** | A daily flood risk forecast for the UK, produced by the FFC (in collaboration with the EA and Natural Resources Wales) to assist with strategic, tactical and operational planning decisions. It gives a flood risk assessment shown by county and unitary authority across |

England and Wales over the next five days for all types of natural flooding (coastal/tidal, river, groundwater and surface water). The FGS is issued by the FFC every day at 10:30am and at other times, day or night, if the flood risk assessment changes.

www.ffc-environment-agency.metoffice.gov.uk/services/FGS_User_Guide.pdf

| | |
|---|---|
| **Flood incident management strategy** | An institute's priorities for preparing for and responding to flood events. |
| **Flood management measures** | Solutions to reduce the impacts that floods pose to humans and the environment. They can be natural (e.g. planting vegetation to retain extra water in the ground) or engineered (e.g. flood barriers). |
| **Flood preparedness** | Measures taken to prepare for and reduce the effects of a flood event. |
| **Flood scenarios** | Possible future development of a flood event and its associated likelihood. |
| **Flood wardens** | Volunteers from local communities who have the responsibility to monitor watercourses in the area they cover and contact local authorities with up to date information. |
| **Forcing** | The action of inputting information into a computer model to produce a forecast. |
| **Forecast accuracy** | The level of agreement between the forecast and the truth (i.e. what is observed in reality). |
| **Forecasting product** | A comprehensive and tailored overview (i.e. in the form of text, graphics and/or tables, etc.) of the forecast. |
| **Hydraulic models** | Mathematical models of the movement of water in a system (e.g. a river). |
| **Hydrological model** | Simplified model of a real-world system that describes the water cycle. |
| **Hydro-meteorological observations and forecasts** | Hydro-meteorology is a branch of meteorology and hydrology that studies the transfer of water and energy between the land surface and the lower atmosphere. Hydro-meteorological observations include observations of meteorological (e.g. temperature and rainfall) and hydrological variables (e.g. river and groundwater levels). Hydro-meteorological forecasts are forecasts that predict the evolution of meteorological and hydrological variables in time. |
| **Hydro-Meteorological Services** | Hydro-meteorological forecasting* products* produced by the FFC and issued daily (Hydro-Meteorological Guidance), twice daily (Forecast Meteorological Data) or whenever required (Heavy Rainfall Alerts). |
| **Lead time** | The length of time between when the forecast is made and the occurrence of the event (e.g. flood) being predicted. |
| **Long-range forecasts** | Forecasts which cover a period of time from a month to more than a season. |
| **Missed events** | An event, for example a flood, for which no warning was given ahead of it happening. |
| **Model performance** | The level of agreement between the model's outputs and their observations in reality. The difference between a model output and its respective observation is the error. The lower the error, the greater the model performance. |
| **Nowcasting** | Extrapolating from the latest observations (e.g. radar rainfall) to forecast the evolution of, for example the weather, in the next couple of hours. |
| **Operational decision-making** | Decision-making based on real-time information to resolve imminent situations. |
| **Outlooks** | Refers to forecasting products* based on long-range forecasts* (i.e. monthly to seasonal). |
| **Performance measures** | Metrics that characterise the quality of a forecast or a model compared to observations. |
| **Probabilistic forecasts** | While a deterministic model gives a single possible outcome for an event, a probabilistic model gives a probability distribution as a solution, indicating the likelihood of each |

| | |
|---|---|
| | scenario to occur. By design, probabilistic forecasts display the uncertainty in our estimates of future water levels, for example. See 'Ensemble' for information on how these forecasts can be produced. |
| **Rapid-response catchments** | Catchments and rivers that respond quickly to rainfall events. |
| **Reasonable Worst Case** | A forecaster's assessment of the potential upper range of rainfall, river and groundwater levels, and coastal conditions, and their impacts. |
| **Risk** | A combination of likelihood and impact of an event. |
| **River level correlations** | Mathematical characterisation of the river level at one point of the river with respect to another point on the river. This can be used to estimate the river level at a point on the river if the river level upstream is known. |
| **Surge forecasts** | Forecasts of the rise of water along coastlines. |
| **Uncertainty** | Having limited knowledge or understanding of our environment, it is impossible to characterise and predict its evolution with 100% certainty. Ensemble* or probabilistic forecasting* can be used to represent the uncertainty in our estimates of future water levels, among others. |

**Appendix B.** Visual examples of current operational forecast products and fluvial flood forecast messages and products used and produced by EA MFDOs and FWDOs for various dates in the past: (a) Flood Guidance Statement, (b) Hydro-meteorological Guidance, (c) Forecast Meteorological Data, (d) Rainfall Scenario Map, (e) and (f) Heavy Rainfall Alert, (g) 'Best Estimate' and 'Reasonable Worst Case' rainfall and water level scenarios on the NFFS, (h) written description of the two forecast scenarios from an MFDO, (i) EA flood alerts and warnings to the public, and (j) fluvial flood alert message to the public. Sources: Fig. (a)-(h) were obtained from the EA, and Fig (i) and (j) were obtained from Twitter and the EA active warnings website: https://flood-warning-information.service.gov.uk/warnings, respectively. Note that these examples are not for the same dates as these were not available. The works published in this journal are distributed under the Creative Commons Attribution 4.0 License. This licence does not affect the Crown copyright work, which is re-usable under the Open Government Licence (OGL). The Creative Commons Attribution 4.0 License and the OGL are interoperable and do not conflict with, reduce or limit each other.

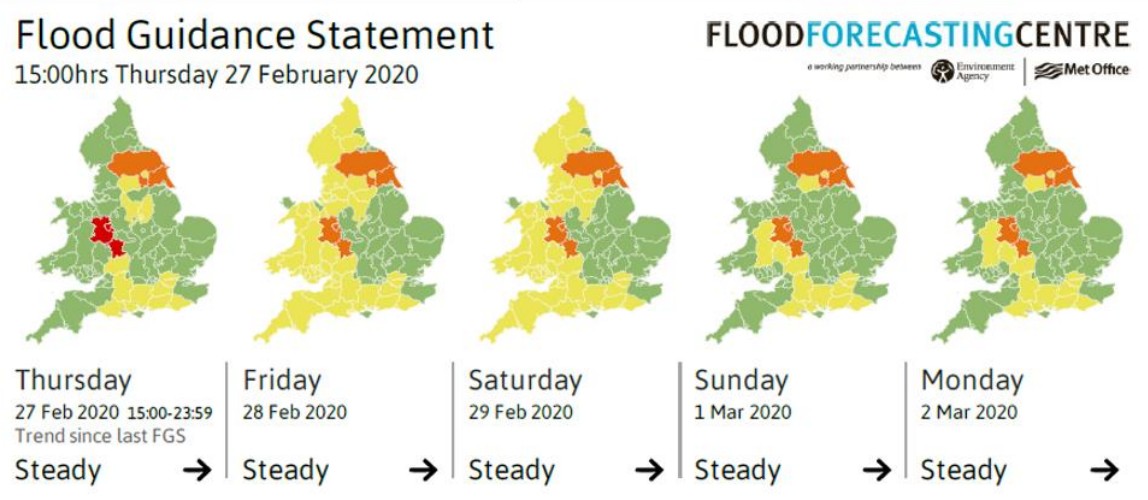

# Flood Guidance Statement
15:00hrs Thursday 27 February 2020

**FLOODFORECASTINGCENTRE**
a working partnership between ⊕ Environment Agency | Met Office

| Thursday | Friday | Saturday | Sunday | Monday |
|---|---|---|---|---|
| 27 Feb 2020 15:00-23:59 | 28 Feb 2020 | 29 Feb 2020 | 1 Mar 2020 | 2 Mar 2020 |
| Trend since last FGS | | | | |
| Steady → | Steady → | Steady → | Steady → | Steady → |

Ongoing severe flooding impacts on the River Severn today becoming significant from Friday. Ongoing significant flooding impacts on the lower River Aire. Further rain from Friday may lead to new flooding impacts. The overall flood risk is HIGH.

## Specific Areas of Concern Map 1: Thursday 27 February 2020

**RISK AREA A**
Impact **SEVERE**
Likelihood **HIGH**

Source  River
Likely duration  1 Day

Severe flooding impacts at Ironbridge, Bewdley and Worcester today (Thursday).

**RISK AREA B**
Impact **SIGNIFICANT**
Likelihood **HIGH**

Source  River
Likely duration  5 + Days

Significant river flooding impacts for the Lower Aire washlands next five days.

**RISK AREA D**
Impact **SIGNIFICANT**
Likelihood **LOW**

Source  River
Likely duration  5 Days

Ongoing river flooding impacts on the Severn.

**RISK AREA E**
Impact **SIGNIFICANT**
Likelihood **LOW**

Source  River
Likely duration  5 + Days

Ongoing flooding through York and in the washlands of the Rivers Aire and Ouse

**RISK AREA F**
Impact **MINOR**
Likelihood **HIGH**

Source  River
Likely duration  1 Day

Ongoing flooding on the River Trent

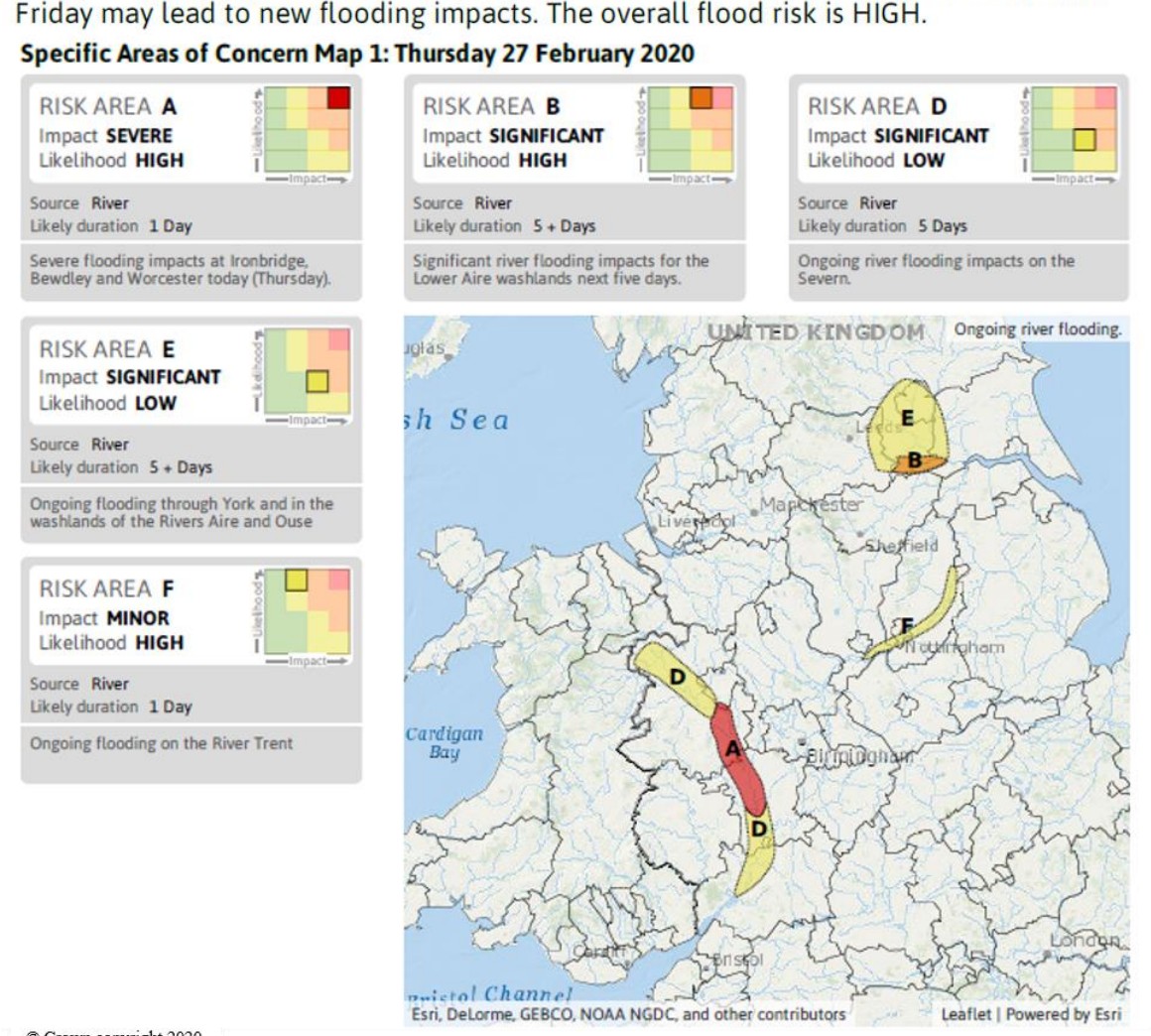

Ongoing river flooding.

## (b) Hydro-meteorological Guidance

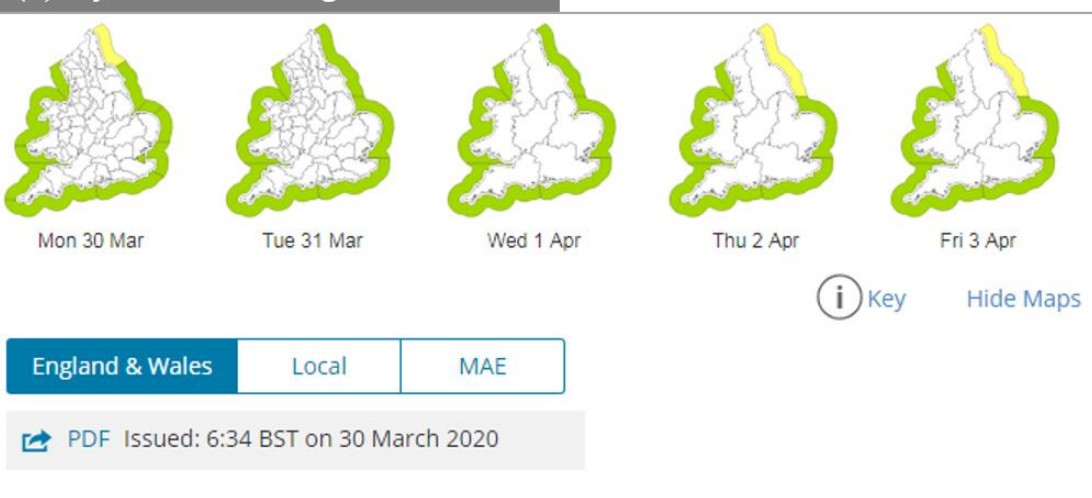

| Mon 30 Mar | Tue 31 Mar | Wed 1 Apr | Thu 2 Apr | Fri 3 Apr |

ⓘ Key    Hide Maps

**England & Wales** | Local | MAE

➦ PDF  Issued: 6:34 BST on 30 March 2020

### Headline

No increased flood risk expected over the next five days.

### Days 1 to 5

Scattered showers move quickly across from North Sea coasts today in a strong north-easterly wind, these wintry over high ground. Showers tending to ease overnight to give a mainly dry day on Tuesday. Wednesday to Friday will then see some patchy light rain and drizzle in parts of the west and north of England and Wales, becoming increasingly wintry over high ground into Friday.

© Crown copyright 2020

## Forecast Meteorological Data
### EA North East Region

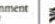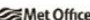

**FLOODFORECASTINGCENTRE**
*a working partnership between* Environment Agency | Met Office

Issued by the Flood Forecasting Centre on 07/11/19 at 07:44 GMT (07:44 local time)
Unique Reference No. 5918 Version 1  Morning Issue

**Precipitation Forecast Days 1 and 2**

| | | Thursday 07/11/19 | | | | | Friday 08/11/19 | | | |
|---|---|---|---|---|---|---|---|---|---|---|
| | | 00-06 (GMT) | 06-12 | 12-18 | 18-24 | Day 1 Total (06-24) | 00-06 | 06-12 | 12-24 | Day 2 Total (00-24) |
| CVHT | Ave (mm) | | 4 | (1) 2 | 2 | (7) 8 | (0) 1 | 2 | 0 | (2) 3 |
| | Max (mm) | | (6) 7 | (3) 4 | (3) 6 | (10) 13 | (2) 3 | (4) 5 | 1 | (5) 7 |
| WPN | Ave (mm) | | (13) 8 | (12) 8 | (6) 3 | (31) 19 | 1 | (1) 2 | 1 | (3) 4 |
| | Max (mm) | | (19) 18 | (24) 20 | (16) 8 | (59) 45 | 2 | (4) 5 | 3 | (7) 8 |
| CNPN | Ave (mm) | | (13) 11 | (7) 5 | (5) 3 | (25) 19 | 1 | (2) 3 | (2) 1 | 5 |
| | Max (mm) | | (21) 20 | (21) 17 | (16) 10 | (57) 47 | 3 | (5) 7 | (5) 3 | 9 |
| SPN | Ave (mm) | | (18) 10 | (17) 15 | (12) 10 | (47) 35 | (6) 5 | 0 | (0) 2 | (6) 7 |
| | Max (mm) | | (23) 17 | (26) 25 | (22) 20 | (67) 55 | (11) 10 | 1 | (1) 5 | 12 |
| NEC | Ave (mm) | | (4) 3 | (1) 0 | (1) 0 | (6) 3 | (1) 2 | (2) 3 | 1 | (4) 6 |
| | Max (mm) | | 6 | 2 | 2 | (9) 8 | (3) 4 | (5) 6 | (4) 2 | (8) 9 |
| MOOR | Ave (mm) | | (6) 3 | (8) 5 | (12) 9 | (26) 17 | (4) 3 | (3) 4 | (1) 3 | (8) 10 |
| | Max (mm) | | (9) 5 | (15) 10 | (20) 16 | (41) 29 | (10) 7 | 7 | (3) 7 | (17) 19 |
| | Ave (mm) | | (9) 7 | (15) 10 | (14) 10 | (38) 27 | (6) 5 | (3) 1 | (1) 2 | (10) 8 |

**Daily Summary Days 1 – 5**

| | | Thursday 07/11/19 | Friday 08/11/19 | Saturday 09/11/19 | Sunday 10/11/19 | Monday 11/11/19 |
|---|---|---|---|---|---|---|
| Precipitation | Ave(mm) | See table above | | (4) 6 | (3) 2 | (6) 2 |
| | Max(mm) | | | (7) 10 | (10) 5 | (13) 5 |

| | | Thursday 07/11/19 | Friday 08/11/19 | Saturday 09/11/19 | Sunday 10/11/19 | Monday 11/11/19 |
|---|---|---|---|---|---|---|
| Temperature | Min(degC) | 0 | -4 | (-5) -6 | (0) -2 | (0) 1 |
| | Max(degC) | 10 | (10) 11 | (9) 8 | 11 | (10) 9 |

## (d) Rainfall Scenario Map

**Rainfall scenario map**

Valid from: 0600 Friday 25 October 2019
Valid until: 1500 Saturday 26 October 2019

**Met Office**

Issued: 0700 Wednesday 23/10/2019

### Area A
**Reasonable Worst Case (very low conf)**

| Widespread | 70 mm | 30 hrs |
|---|---|---|
| | 40 mm | 6 hrs |
| Locally | 100 mm | 30 hrs |
| Isolated | 200 mm | 30 hrs |
| | 70 mm | 6 hrs |

### Area B
**Reasonable Worst Case (very low conf)**

| Widespread | 50 mm | 30 hrs |
|---|---|---|
| | 30 mm | 6 hrs |
| Locally | 70 mm | 30 hrs |
| Isolated | 160 mm | 30 hrs |
| | 60 mm | 6 hrs |

### Area C
**Reasonable Worst Case (very low conf)**

| Widespread | 30 mm | 30 hrs |
|---|---|---|
| | 15 mm | 6 hrs |
| Locally | 40 mm | 30 hrs |
| Isolated | 80 mm | 30 hrs |
| | 40 mm | 6 hrs |

**Rainfall spatial definitions**
**Widespread:** Greater than 60 % of the main area.
**Locally:** Approximately 20-40 % of the main area.
**Isolated:** Less than 20 % of the main area.

**Dashed sub-areas** ·········
These give an indication of where the 'Locally'
and 'Isolated' totals are more likely to apply.

All times are local unless stated otherwise

**Background map: Issues with production tool.**
Best Data Medium Range
22/10/19 1800 GMT represents
the BE scenario for the period.
See figure 2 for 24hr
accumulations.

## Heavy Rainfall Alert
### EA North East Region

**FLOODFORECASTINGCENTRE**

a working partnership between  Environment Agency | Met Office

Issued by the Flood Forecasting Centre on 06/11/19 at 18:21 GMT (18:21 local time)
Unique Alert Reference No. 2922_NORTHEAST_1117   Version 2

**UPDATE**

Start of meteorological event: 0300 GMT on 07/11/19

End of meteorological event: 0600 GMT on 08/11/19

**Summary of Alert Criteria Met**

| Alert Criteria | HRA Areas covered | Confidence |
|---|---|---|
| 10 mm (or more) in 1 hours (or less) | North East Coast, West Pennines, Central & Northern Pennines, South Pennines, Moors, Vale & Wolds | L |
|  |  |  |
|  |  |  |
|  |  |  |
| 15 mm (or more) in 6 hours (or less) |  |  |
|  |  |  |
|  | North East Coast, West Pennines, Central & Northern Pennines, South Pennines, Moors, Vale & Wolds | H |
|  |  |  |
| 40 mm (or more) in 12 hours (or less) | North East Coast, West Pennines, Central & Northern Pennines, South Pennines, Moors, Vale & Wolds | M |
|  |  |  |

*Notes:*
- **Confidence:** *The probability of this threshold being achieved anywhere in the specific HRA Area within the time periods outlined by the Heavy Rainfall Alert. H = more than 60%; M = 40 – 60%; and L = 20 – 40%*
- *Issue of a Heavy Rainfall Alert means the probability of rainfall thresholds being met or exceeded during the meteorological event is within the bands indicated by the confidence levels above.*
- *All Alert criteria should be defined in this table. If it is predicted that some criteria will not be exceeded, these boxes should be greyed out*

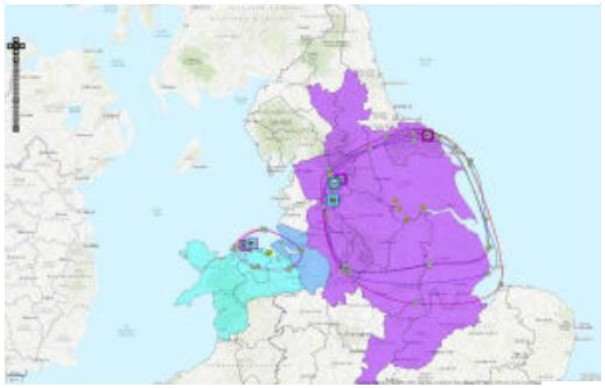

## (f) Heavy Rainfall Alert

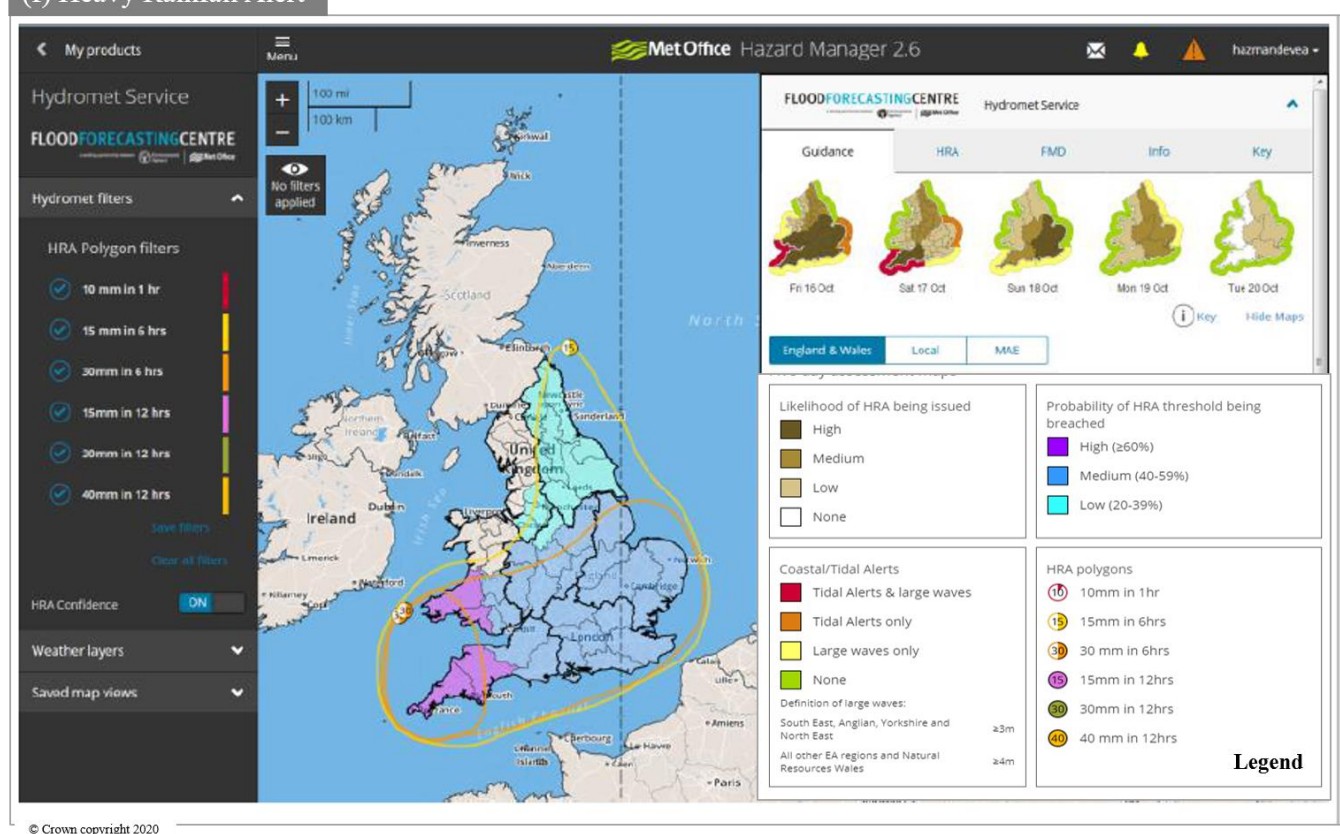

© Crown copyright 2020

## (g) BE and RWC scenarios on the NFFS

### Darton

P.merged [1] (FMRBE)
P.merged [2] (FMRRWC)
P.snowmelt.merged [1] (BE)
P.snowmelt.merged [2] (RWC)
P.snowmelt.merged [3]

H.obs
H.rated.hist.upd [3] [PDM]
H.rated.fcast.upd [1] [PDM] (BE)
H.rated.fcast.upd [2] [PDM] (RWC)

© Crown copyright 2020

*"The rain fall over Saturday night didn't cause much response on the upper catchments. With this we are not expecting a significant increase in the lower catchments (including the Aire).*
*Based on current forecasts:*
- *Based on the BE does not see Carlton Bridge reaching the 4.4 –4.5m stage we do not expect to see it start to fill the West Marsh Washlands.*
- *Based on the RWC (with a +3 degree Celsius increase to account for snow melt) we do not expect to reach the 4.4 –4.5m stage @ Carlton Bridge*

*The forecast for the coming week is as follows: Light wintery showers feeding in from the west in the next couple of days. Tuesday –further signals of showers for 5-10mm of rain / snow over the high grounds. This is represented in the river forecast @ Carlton Bridge, however levels will remain below 4.4m Stage. Wednesday to Thursday –further low pressure is expected, however confidence is extremely low and currently expected to impact south of the country."*

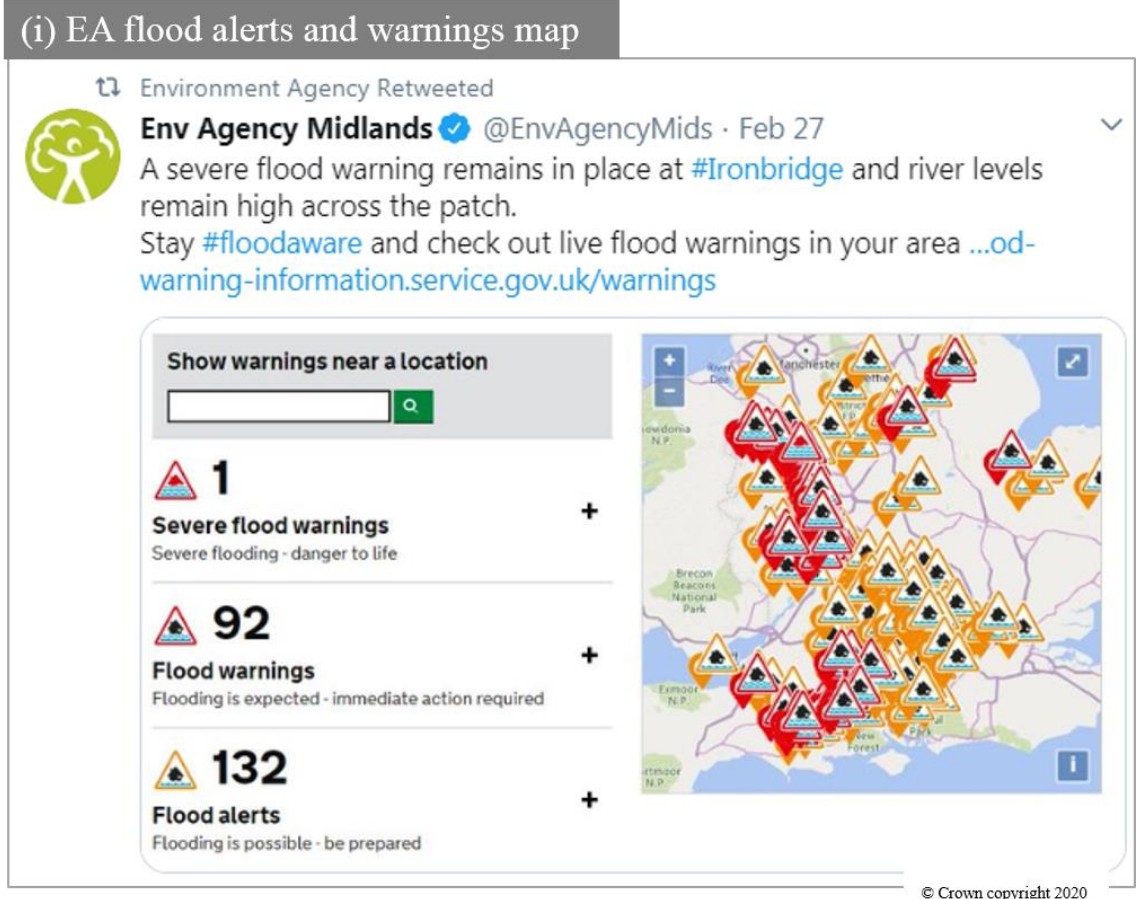

# River Lambourn and its tributaries from Upper Lambourn down to Newbury

**Flooding is possible - be prepared**

Property flooding is not currently expected. River levels are high but stable on the River Lambourn as a result of high groundwater, particularly in the Eastbury area where the flood control structure continues to operate. Levels are high on the Winterbourne at Bagnor where the river is still out of banks and will be sensitive to further rainfall. The risk of flooding of low-lying land, footpaths and roads remains. Please remain aware of your local surroundings and stay safe by avoiding contact with flood waters. The forecast is for dry weather with some scattered showers until Friday. Please refer to the 'River and Sea levels in England' webpage for current river levels and remain safe and aware of your local surroundings. This message will be updated tomorrow, 01/04/20, or if the situation changes.

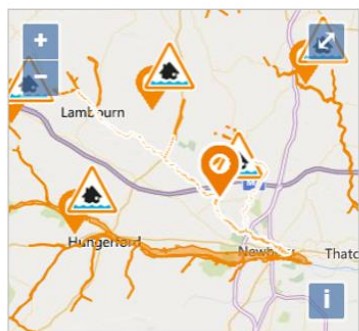

Hide other warnings and alerts

This information was last updated at **10:53am Tuesday 31 March 2020**