# Peer review of ""Are we talking just a bit of water out of bank? Or is it Armageddon?" Front line perspectives on transitioning to probabilistic fluvial flood forecasts in England"

_Geoscience Communication, 2019_

## Short Comment (SC1) · 8 Oct 2019

This paper provides a detailed overview of the role of FWDOs and MFDOs within the EA which as far as I'm aware isn't currently available within the academic literature. It also provides a timely discussion on the practical perspectives of transitioning to probabilistic fluvial forecasts within the EA supported by rich interview evidence. The combination of these two aspects provides a valuable contribution to the field in a well presented paper.

[Figure]

On reading the paper there are a couple of areas that I think could be clearer, particularly around the definitions of lead times and decision makers.

1. Firstly the importance of timings/lead time is confusing as the focus changes throughout the paper from the 2-6 hour window for issuing flood warnings to the 2-5 day window for more strategic planning decision making. The relative value, and expected use of probabilistic forecasts will be different at these two timescales. The quote on line 270 is key to this discussion. It would be helpful to have a clearer focus on what lead time is being discussed at different points in the paper. A specific example is line 234 talking about waiting for the forecast to be confident – I expect there is also a balance of confidence and lead time here

2. Largely the paper considers the FWDOs as the decision makers as they are the ones issuing flood warnings. The latter part of the paper also introduced external stakeholders as decision makers, making decisions on the EA flood warnings. It would improve clarity to make it clearer when you are talking about internal and external decision makings. In particular this applies to the discussion around lines 500 and 540. Maybe also consider adding a definition of 'decision maker' to the glossary and recognising that decisions are made at various points during the forecasting chain.

3. Similarly it is not clear who makes the decisions re increasing control room staffing or increasing field resources (e.g. for putting up flood defences). Is this the FWDO, the MFDO or someone else? And what time scale are these decisions made on/how will the change to probabilistic forecasts affect them?

4. I know it is beyond the scope of the paper but there is limited consideration of how the change to probabilistic forecasts might affect external decision makers although it is raised as a concern by FWDOs. A useful extension to the work would be to see what information those receiving flood warnings actually want, to understand if it is a perceived or real concern about the lack of binary forecast information and the pushing of decision making further down the chain. This is also relevant to recommendation 1

– as well as communicating that there will be a change, should the EA also have some responsibility for preparing external decision makers on how this might change their practices?

The following are some minor text edits:

Line 19 – I would remove 'alternative', I'm not sure why these two areas are alternative.

Lines 130:134 – this is largely a repeat of lines 76:81.

Line 224 – increase/ change of 200%?

Personally figure 5 (the wordcloud) doesn't add much to the paper for me.

Given the quotes listed in the appendix, I think the wording of table 1 could be stronger.

---

## Referee Comment (RC1) · Erin Coughlan de Perez (Referee) · 15 Oct 2019

This paper is a fascinating glimpse into the day-to-day operations of flood forecasters. It is very helpful to have this described in the published literature, to understand how information is communicated and how forecasters interpret model results before issuing alerts.

While the concept of the paper is excellent, my major suggestion is to re-work the framing. As a reader, it is difficult to follow the logic of the paper. Often, it seems to be

an interesting mix of quotes from interviews, lacking some analysis to identify relevant points and present them in a structured manner.

The introduction provides interesting context to the flood forecasting situation in the UK, but the details often seem disconnected. It is not quite clear how the authors are leading up to a solid research question. At the end of the introduction, it becomes clear that the EA is going to transition to probabilistic forecasts, and that the researchers will interview the forecasters about this transition. However, it is not really clear what the purpose is for the interviews. My suggestion is to re-write the introduction with a clarified mandate, leading up to the question that will be then answered in the paper.

Similarly, the section 4 is very interesting; it is neat to have a window into the ways of working of these flood forecasters. However, the quotes and text do not form a coherent story for the reader. The section 4 does not seem well linked to the recommendations in section 5 – it is not clear to the reader how the interviews resulted in the recommendations. They read more like a policy brief than a research output; perhaps this would be better suited as a commentary than a research article?

In particular, section 5 seems to bring in the idea of decision-makers, but all of the interviews were with forecasters and those responsible for issuing warnings. There was no analysis in section 4 of who the decision-makers are, what decisions would be made based on the warning, or how that affected the information that was released.

The recommendations in section 5 seem to be generally good practice recommendations for whenever a forecasting system might have some sort of change, but not necessarily linked to the research results. In addition, most of the recommendations do not seem to have any relation with the characteristics of this particular change in forecast systems; the fact that the new system is probabilistic rather than deterministic.

Here are some examples of additional places where the text could be changed to present a clear research statement to the reader: Section 3.3 question 2 – as a reader first encountering this statement, I am not sure what you mean by this. Section 3.3
question 3 – when you say "potential impacts", what does this mean? As a reader, I can't anticipate what exactly you are looking for.

The text would benefit from a few additional examples, if possible.

Some additional points for clarification: Page 6 line 213: How is running the NFFS different from the information given by the FFC? How will those NFFS localized model runs update or change the flood forecast of the FFC? It would be helpful to have a worked example, of what was produced by the FFC, what was communicated to the forecasters, what additional data they gathered, and what warnings they then communicated to people and what decisions were made based on that information.

Page 6 line 234: What do you mean that they wait for the forecast to be "confident"? Is this not a deterministic forecast?

The title is eye-catching, but the question that is being asked by the interviewee (water over bank vs. Armageddon) is about the magnitude of the event rather than about probabilities, while the paper is about transitioning to probabilistic forecasts.

---

## Referee Comment (RC2) · Anonymous Referee #2 · 25 Oct 2019

Dear authors,

Thank you for the interesting read. To me this is a valuable insight in the reality of flood forecasting and (early) warning in practice, provided first hand, by the duty officers of the UK Environment Agency. I appreciate the choice of the authors to present many quotes from the interviews performed, which helps in their attempt to make an authentic account of the current practice, as well as the expectations of the upcoming introduction of a probabilistic flood forecasting system.

[Figure]

I have the following general comments:

The experience with the recent (2013/2014) transition from single forecast to 2-scenarios may be quite relevant for the perception of the interviewees and prospects of the upcoming change from two scenarios to probabilistic forecasts. Did you discuss this with the interviewees and could you perhaps elaborate more on this aspect in the paper?

From reading the paper I get the impression that there is little known yet about what will be the procedures for preparing, communicating internally, and use in warning decision making of the probabilistic forecasts. Could you, for example in the Context section, elaborate on what is known, and what was known to the interviewees at the time of the interviews, about the upcoming transition to probabilistic forecasts? If nothing is known yet, the good thing is that the recommendation of co-design (recommendation 3) can still be taken up, and at the same time it might explain some of the perceived challenges associated with the upcoming transition to using probabilistic forecasts.

The presentation and discussion of the answers of the interviewees to the last question, about the upcoming introduction of probabilistic flood forecasts, seems to me to be somewhat limited. This impression is fed by the sudden change in reporting format from in-line quotes to a wordcloud, summarising table (Table 1), and a reference to an Appendix, none of which are discussed in the manuscript text (lines 460-464). I may be overlooking something, and if not, you may well have chosen this approach for good reasons. If by design, then I would recommend to explain the reasons in the same section (Section 4.3.2). If possible, however, I would recommend continuing with the reporting format of the previous sections, or at least including a discussion of the wordclouds and Table 1.

Also the Discussion and Recommendation section leaves me with a feeling that more reflection on the interview results can be done. It would be, for example, interesting to reflect on whether the interviewees' answers to the first questions are in-line with,

help explain, or not, their perceived opportunities and challenges (last question). The 10 recommendations in section 5.2 seem somewhat disconnected from the interview results (only Refs to literature are given). I would recommend to put in, in Section 5, more references to findings reported in earlier sections of the paper.

Specific questions and comments:

- You focus on the benefit of probabilistic forecasts of increasing lead time (e.g. p1 l14 and l36, p3/4 l124/125, while other benefits include the potential of increasing the probability of detection of floods (reducing missed events), and supporting risk-based decision making. Could you reflect on this in the text? E.g. adding advantages or explaining why you refer mainly to increasing lead time. Interview findings indicate that in the current practice final decision of issuing a warning is often based on nowcasts with lead times of only a couple of hours and/or on observations (e.g. page 7 line 267 and page 11 line 397/398). Do the interviewees and/or you think that the introduction of probabilistic forecasts is going to change this practice? Could you reflect on this in the paper?

-Some of the duty officers (DOs) seem to be concerned about how the probabilistic forecasts will be received by the action response units. Does this mean that in the current practice, forecast hydrographs are send along with the warning to action response units? Please clarify in the text.

- I may have missed it, but could you include more information (and refs if available) on the probabilistic forecasting system that will be used? Is it based on meteorological ensembles or on another probabilistic forecast method? Will the MFDOs be responsible of running it through the hydrological models (as they seem to be now)? Will hydrological uncertainty also be included in the hydrometeorological ensemble, and how? etc. Please also reflect on whether this information on the features of the new forecasting system was known to the interviewees.

-When reading Table 1, I do not perceive a strong concern about the upcoming intro-

Interactive
comment

duction of probabilistic forecasts, while when reading the quotes of Annex C I do sense a strong concern among the Duty Officers interviewed. This concern seems mainly to be that probabilistic forecasts will put all the responsibility of taking a decision with themselves (rather than with the forecasters or with the action response units). Could you note and discuss this in Section 4.3.2.? And then elaborate on recommendations on how to prevent/manage that? For example, Recommendation 9), setting guidelines on '..the forecast confidence at which certain decisions and actions should be made..', may also not be the answer, because the DOs indicated in the present-day practice the value of local expert judgement in issuing warnings and seem to appreciate having the freedom of applying such expertise. Prescribing decision making rules, may, therefore, be a step too far in taking away forecast interpretation responsibility from them.

- Could you reflect on whether the interviewees see the probabilistic forecasts as an additional input to their flood forecast confidence assessment? The DOs are already communicating a confidence level with the warnings they send-on down the line. One quote in Appendix 3 confirms seeing this as an opportunity, but it would be interesting to read from you what is your impression on this for the other interviewees.

- Page 4 line 130/131 refers to the EA already using probabilistic coastal flood forecasts. Why not learn from the experiences of that earlier transition (perhaps too long ago), or at least from the user experiences. Could you elaborate a bit more, e.g. whether or not you think that would be interesting for other researchers and the EA to pick up.

- Page 5 lines 182-184: Consider referring back to these research questions in your Conclusions section.

- Page 6 lines 222-227: Choosing the What-if scenario could be perceived as quite a responsibility. A responsibility that might be (partly) taken away with the introduction of probabilistic forecasts. Did you discuss this with the MFDOs and what are their and your thoughts on this? Consider elaborating on this in the paper.

- Are all the warnings issued being archived (including alerts, issue time, updates, etc.)? Are actual flood occurrences being documented? And if so, are the archives being compared and analysed? Could you reflect on this, and do you think such analysis could be/has been helpful for identifying challenges in the current forecasting system and warning practices, as well as for analysing in the near future the impact (or lack thereof) when introducing the probabilistic forecasts and identifying persistent and potentially new challenges?

- Page 11 line 391/392: Could these differences perhaps also be a consequence of differences in catchment size/rainfall-runoff response time/land use and differences in flood management actions that follow the warnings and the time these measures take? Consider mentioning/reflecting on this (at this point) in the paper.

- Page 11 line 400: Are the DOs being scored, and if so, how are they scored, and what are these scores used for? If possible, would be interesting to comment on below this citation, and perhaps consider to reflect on how such scoring may have an encouraging or discouraging impact on the uptake of probabilistic forecasts.

- The paper concerns the upcoming transition from a 2-scenario forecast to a probabilistic forecast, but the Supplementary material seems to focus on the recently completed transition from a single forecast to the two scenario's (following 2013/2014). Please clarify, e.g. in the author response, not necessarily in the manuscript.

Detailed comments and editorials:

p1 l12 Consider ..inclusion of uncertainty information in..

p1 l13 Consider ..potential upcoming floods.. instead of 'future' to avoid confusion with climate change. Also consider for other occurrences of 'future'.

p1 l18 Consider ..understand their perception on how this transition..

p1 l24 This sentence is rather broad and in my view not necessary. Consider leaving it out. Instead consider putting some of the key findings and recommendations (similarly

to what is written in the Conclusions section)

p1 l38/39 Consider ..given the explicit provision of uncertainty information.. Single forecasts are as uncertain as probabilistic forecasts. The uncertainty is just not shown.

p2 l42/43 ..designed to capture scenarios that may not always realise.. That does not sound quite right/the point of probabilistic forecasting. Consider just leaving that whole sentence out, or reformulate to something like "Warning on the basis of low probabilities of flood, for example, will reduce the chance of missing an event, but will also lead to more false alarms."

p2 l44 Consider .. when a pre-defined threshold (e.g. river stage) is reached..

p3 l88 Consider ..whilst local flood authorities..

p3 l113 I would suggest including here an explanation on how the two scenarios are prepared. (I realise this is described later, but it left me curious from this point onward, especially because it matters for the context of the interviews to what extend these two scenarios are a step towards probabilistic forecasting or not).

p3 l114-118 I am not sure if I now understand correctly how the two scenarios should be used. This might be due to me not being a native speaker, but if you could reformulate to further clarify, that would be appreciated.

p3 l120 ..potential risks of impacts.. Please reformulate.

p4 l129 I do not think you can 'ensure' the appropriate use. Consider to reformulate, e.g. 'support'.

p5 l171 ..advance..

p5 l186 ..communicated by several interviewees,..

p6 l234 'waiting for the forecast to be confident' Please explain how the forecast can become confident in the context referred to here (present practice).
p7 l257 please add who has the 'Expert knowledge'.

p7 l264 please also describe how/for what/when the 'reasonable worst case' should be used.

p9 l345 please consider to add also the procedure for using the reasonable worst case again.

P10 l380 - 383 seem to me a bit too personal. Kindly double-check.

p12 l453 In my view this is an important finding that can also be used in the upcoming transition (and gives reason for a positive prospect). I cannot recall whether you clearly refer back to this finding in the Discussion and Conclusion sections at the end of the paper, but if not, I recommend including it.

p13 l463/464 Unclear, and appears as a stand-alone sentence. I suggest making this part of a discussion to be added, of Fig. 5, Table 1, and Appendix C. (See also my third general comment)

p13 l463 consider ..sound extreme.. or alternative formulation.

p13 l469/470 I do not understand this sentence. If we achieve increased confidence levels before moving up lead time, why is the second part of the sentence, about the chaotic system, posing problems? Consider clarifying or leaving out the sentence.

p13 l472/473 ..'uncertain' science.. Not sure I understand. Do you mean new discoveries being at first 'uncertain' (or not trusted), until the experiment has been reproduced with the same results (or tested in pilots, practice, etc.)? Or do you mean, more specifically, the science of quantifying uncertainty associated with predictions? Consider reformulating.

p13 l490/491 consider ..decision makers operate, and where the forecast..

p13 l497 consider ..uncertainty information.. or ..information on uncertainty..

[Figure]

p14 l512/513 ..crucial to develop a methodology.. Not sure if a single methodology is the 'crucial' solution of the challenge of using probabilistic forecasts. It may be that case-dependent and user-community dependent ways have to be found, by scientists and users together, on how to effectively use confidence (uncertainty) information.

p14 l418 I do not think this reflects the main idea of probabilistic forecasts, nor that the provision of scenarios causes more false alarms. I would argue the other way around, that having the scenarios, the probabilistic forecasts, gives the opportunity to the decision maker (and its beneficiaries) to balance the number of missed events and false alarms to their needs. Please reformulate.

p14 l425 This seems a bit out of the blue and not clear to what extend you think this relates specifically to EA communication pathways and warning procedures.

p14 l531 Consider adding a brief explanation on what is a 'post-factual society'.

p15 l547 consider ..we made a list..

p15 l548 consider something like ..The recommendations concern actions we think the EA should take with high priority..

p15 recommendation 1: Before this campaign, should there first be known a bit more on how the change will be done, or not?

p15 recommendation 2: consider ..to all players..

p15 l577 is this recommendation specifically for EA, or for the flood forecasting and early warning research community. If the latter, consider moving this elsewhere.

p16 l582/583 given the reported differences in catchments, forecast performance, impacts, and warning response actions, etc., double-check this recommendation. Consider 'customised' rather than 'homogeneous'.

p16 l601 consider ..should be collected and used to update design and procedures..
p16 l602 consider ..To handle situations..

p16 l613 stand-alone sentence, consider moving somewhere else or elaborating.

p17 l623 because of the mentioned concern of the responsibility of decision making being pushed down to the DOs, I would not write 'lie mostly outside their role', rather just something like 'the main perceived challenges concern..'. In the sentence before, perhaps it would be more clear to write something like '..concerns about impacts of this transition on the communication and interaction between them.'

p17 l624 consider replacing 'translating uncertain information to a binary decision' (because it is a challenge that they already perceive in the present situation) for the worry of the responsibility of decision making being pushed further to them.

p17 l625 consider reformulating to something like ..High priority actions were recommended to the EA.. to support a successful..

p23 Caption figure 4: consider ..Complex flood forecast interpretation landscape.. because the decision making landscape includes more elements, such as external pressure.

p23 Caption figure 5: please add from what the opportunities and challenges arise (e.g. ..from the introduction of probabilistic flood forecasts)

p24 Title table 1: consider ..A sample of supporting quotes..

p24 Table 1 line 2: 'improve long-term communication' please clarify.

p24 Table 1 line 2: add a full-stop at the end of the sentence.

p24 Table 1 line 5: consider ..contain information on forecast uncertainty..

p24 Table 1 line 22: ..it is worth noting that..

---

## Referee Comment (RC3) · Jan Verkade (Referee) · 10 Nov 2019

The manuscript describes the 'transition' of the Environment Agency towards using probabilistic fluvial flood forecasts. The topic is very relevant and worthy of analysis indeed.

Having said that, I think the manuscript would benefit from (i) a better description of what it aims to achieve (ii) additional analysis that would justify the recommendations that are made.

[Figure]

Re (i). The research, through the interviews, provides data that documents the view of various individuals at a single point in time, regarding a change in an organization's processes and procedures. I like to make the analogy with "observations" in the quantitative sciences. Observations can be used to provide evidence of the plausibility of some hypothesis - or of the absence thereof. Such a 'hypothesis' element would greatly improve the quality of the manuscript - and provide a response to the "So What?" question that, post reading the manuscript, continues to linger in my mind.

Re (ii). Based on the interview data, recommendations are phrased. While some of these may be very worthwhile indeed, I think recommendations can only be made based on an analysis where objectives are offset with achievements or projected achievements. Ergo, I think recommendations can only be made if the agency's objectives (with respect to the production and use of probabilistic forecasts) are described. Have these objectives been described in the 2016 National Flood Resilience Review that is cited, maybe? Or are documented elsewhere?

Please also note the supplement to this comment:
https://www.geosci-commun-discuss.net/gc-2019-18/gc-2019-18-RC3-supplement.pdf

**Supplement:**

**Review of *"Are we talking just a bit of water out of bank? Or is it Armageddon?" Front line perspectives on transitioning to probabilistic fluvial flood forecasts in England* by Louise Arnal et al.**

Jan Verkade, Delft, October/November 2019

**Overall impression**

The manuscript describes the 'transition' of the Environment Agency towards using probabilistic fluvial flood forecasts. The topic is very relevant and worthy of analysis indeed.

Having said that, I think the manuscript would benefit from (i) a better description of what it aims to achieve (ii) additional analysis that would justify the recommendations that are made.

Re (i). The research, through the interviews, provides data that documents the view of various individuals at a single point in time, regarding a change in an organization's processes and procedures. I like to make the analogy with "observations" in the quantitative sciences. Observations can be used to provide evidence of the plausibility of some hypothesis - or of the absence thereof. Such a 'hypothesis' element would greatly improve the quality of the manuscript - and provide a response to the "So What?" question that, post reading the manuscript, continues to linger in my mind.

Re (ii). Based on the interview data, recommendations are phrased. While some of these may be very worthwhile indeed, I think recommendations can only be made based on an analysis where objectives are offset with achievements or projected achievements. Ergo, I think recommendations can only be made if the agency's objectives (with respect to the production and use of probabilistic forecasts) are described. Have these objectives been described in the 2016 National Flood Resilience Review that is cited, maybe? Or are documented elsewhere?

Maybe the above issues could be resolved through the following:

- There is a document that says that the EA should be moving towards probabilistic forecasting. (NFRR but also the 2008 Pitt review?)
- This overall objective has been adopted by the agency and existing projects/policies (evidence, please!) are in place to try and achieve that objective. As an aside, I wonder if this is indeed the case.
- Specifically, this means that EA will have to do this-and-this ('specific objectives'). If these specific objectives have been described in a policy document, great - use these in your manuscript. If not described, make them up - what could be plausible objectives?
- To meet those specific objectives, the agency will have to do this-and-this. This would be the description of your organizational transition.
- We've gathered some data to try and identify where in that process the EA currently is, what pitfalls they see and where they think the challenges are.
- Offsetting specific objectives versus these 'observations', we note that ... agency is well on its way / straying from its path ... either way, recommendations are...
- It's useful to publish this is in the scientific literature so that (i) scientists may comment on implementation of science in a public organizations; (ii) other organizations may benefit from this; (iii) in assessing the progress of their transition, the EA can benefit from this analysis.

So overall - I think this manuscript is a rough diamond that needs polishing. I am very much interested in seeing the end result - and would be happy to help out through additional reviews if these would be considered helpful.

**Other points**

I think the above would require fairly significant restructuring of the manuscript and I don't think it's worthwhile to, at this stage in the review process, point out any minor issues.

Some scattered observations though:

- The citation in the title does not pertain to the theme of the manuscript. I also find that the manuscript tends to use language that, at times, can be a little more 'dramatic' than required. This is exemplified by the title's "front line perspectives" (a change in processes and procedures is not a war!) and the reference to "Armageddon" (which, by the way, is a settlement on top of a hill - the 'Ar' originates from the Hebrew 'har' which means mountain - and not prone to flooding and hence reference thereto is somewhat unfortunate - but I am digressing now). Additional examples: "the chaotic and far from certain world we live in", "urgently required", "high priority recommendations".
- A glossary is, I think, unnecessary, and I find the asterisks a little distracting.
- The manuscript's theme is the 'transition to probabilistic forecasting'. The amount of text that is dedicated to that theme, however, is relatively small. For example, the Results sections spans lines 190 through 464 - yet only as of line 425 are the probabilistic forecasts discussed. Similar observations can be made to the manuscript as a whole. In my view, the reader is distracted a lot from the main points.
- The language used to describe the somewhat technical aspects of predictive uncertainty could be a little more precise. Some examples:
    - It's **not** forecasts themselves that are uncertain. What's uncertain is the future water levels, streamflows, etc. - also even if an estimate of those future values is made through a forecast. When that residual uncertainty is quantified or expressed, we have available 'estimates of uncertainty', rather than 'uncertainty'.
    - Uncertainty estimates and probabilistic forecasts are not the same thing. Hopefully the level of uncertainty can be expressed as a probability, but very often it cannot. I wouldn't want to use the terms interchangeably in a manuscript.
    - In a manuscript that discussed 'uncertainty', you have to be a little careful with using the word 'certain' ('certain decisions', l609). In most of those cases, the word 'certain' can be either safely omitted, or replaced by 'various'.
- Minor, minor issue: In author contributions, why not simply reference first names? "Hannah, Louise and Susan posed the original question" is a lot easier to read than "H.L.C., L.An. and S.M. posed the original question".

---

## Author Comment (AC1) · 17 Dec 2019

**Authors' responses to interactive comment SC1 from Linda Speight**

This paper provides a detailed overview of the role of FWDOs and MFDOs within the EA which as far as I'm aware isn't currently available within the academic literature. It also provides a timely discussion on the practical perspectives of transitioning to probabilistic fluvial forecasts within the EA supported by rich interview evidence. The combination of these two aspects provides a valuable contribution to the field in a well presented paper.

We thank Linda Speight for her valuable comments which will help improve this manuscript for its final publication.

On reading the paper there are a couple of areas that I think could be clearer, particularly around the definitions of lead times and decision makers.

1. Firstly the importance of timings/lead time is confusing as the focus changes throughout the paper from the 2-6 hour window for issuing flood warnings to the 2-5 day window for more strategic planning decision making. The relative value, and expected use of probabilistic forecasts will be different at these two timescales. The quote on line 270 is key to this discussion. It would be helpful to have a clearer focus on what lead time is being discussed at different points in the paper. A specific example is line 234 talking about waiting for the forecast to be confident – I expect there is also a balance of confidence and lead time here

This is a very good point and we will clarify throughout the paper (where relevant) the lead times-decision types being referred to (from strategic planning up to warning). In fact, we think that this warrants further discussion and distinction in the Discussion and for the recommendations. There is a value to probabilistic forecasts for both strategic planning and warning.

2. Largely the paper considers the FWDOs as the decision makers as they are the ones issuing flood warnings. The latter part of the paper also introduced external stakeholders as decision makers, making decisions on the EA flood warnings. It would improve clarity to make it clearer when you are talking about internal and external decision makings. In particular this applies to the discussion around lines 500 and 540. Maybe also consider adding a definition of 'decision maker' to the glossary and recognising that decisions are made at various points during the forecasting chain.

We will introduce the internal (FWDOs and to some extent MFDOs) and external decision-makers when presenting Fig. 1, making it clear that these interviews and paper focus on internal decision-makers but that external decision-makers also exist. Thereafter in the text we will make sure to clarify which decision-makers we are referring to (such as around lines 500 and 540). We will also add decision-makers to the Glossary.

3. Similarly it is not clear who makes the decisions re increasing control room staffing or increasing field resources (e.g. for putting up flood defences). Is this the FWDO, the MFDO or someone else? And what time scale are these decisions made on/how will the change to probabilistic forecasts affect them?

Decisions regarding an area's response to and preparation for a flood event are made higher up the chain, at the top of the Area Response Unit on Fig. 1. This will be clarified in Section 4.1, when discussing the MFDO and FWDO roles. More information about the timescales of preparations and

actions will be added to Section 4.1 as well. Timescales and how this transition will affect them will be explored in the Discussion and recommendations (as per our response to comment 2. above).

4. I know it is beyond the scope of the paper but there is limited consideration of how the change to probabilistic forecasts might affect external decision makers although it is raised as a concern by FWDOs. A useful extension to the work would be to see what information those receiving flood warnings actually want, to understand if it is a perceived or real concern about the lack of binary forecast information and the pushing of decision making further down the chain. This is also relevant to recommendation 1 – as well as communicating that there will be a change, should the EA also have some responsibility for preparing external decision makers on how this might change their practices?

Although beyond the scope of this paper, this is indeed a very interesting and important point. We will discuss this point more thoroughly in the Discussion, incorporating it into a recommendation and suggesting this as future work.

The following are some minor text edits:

We will include these minor edits.

Line 19 – I would remove 'alternative', I'm not sure why these two areas are alternative.

Lines 130:134 – this is largely a repeat of lines 76:81.

Line 224 – increase/ change of 200%?

Personally figure 5 (the wordcloud) doesn't add much to the paper for me.

We will consider removing the wordclouds from this paper.

Given the quotes listed in the appendix, I think the wording of table 1 could be stronger.

Table 1 will be adapted into text (following a similar format to the other Results sections) and reworded, as per comments from the other reviewers.

---

## Author Comment (AC2) · 17 Dec 2019

**Authors' responses to interactive comment RC1 from Erin Coughlan de Perez**

This paper is a fascinating glimpse into the day-to-day operations of flood forecasters. It is very helpful to have this described in the published literature, to understand how information is communicated and how forecasters interpret model results before issuing alerts.

We thank the reviewer, Erin Coughlan de Perez, for her positive comments with regards to this paper's added value in published literature, as well as her valuable feedback which will help improve the paper for final publication.

1) While the concept of the paper is excellent, my major suggestion is to re-work the framing. As a reader, it is difficult to follow the logic of the paper. Often, it seems to be an interesting mix of quotes from interviews, lacking some analysis to identify relevant points and present them in a structured manner.

As per your comments and comments from other reviewers, the paper's framing will be reworked to provide a more logical storyline. More specifically, the following changes will be made:

- The Introduction and Context sections will be merged into a single section which highlights the EA's current policy and objectives with regards to undertaking/implementing probabilistic forecasts in practice. This will provide a framework for the objectives of this paper (see our response to comment 2) below).

- Section 4.2 will be merged with section 4.1 to provide contextual information about the duty officers' roles (see our response to comment 6) below).

- Section 4.3, one of the highlight sections of this paper, will be rewritten following the same format as the rest of the paper (text and supporting quotes) to constitute a more substantial section.

- The Discussion will be rewritten to link more clearly the interview results and recommendations (see our response to comment 3) below).

2) The introduction provides interesting context to the flood forecasting situation in the UK, but the details often seem disconnected. It is not quite clear how the authors are leading up to a solid research question. At the end of the introduction, it becomes clear that the EA is going to transition to probabilistic forecasts, and that the researchers will interview the forecasters about this transition. However, it is not really clear what the purpose is for the interviews. My suggestion is to re-write the introduction with a clarified mandate, leading up to the question that will be then answered in the paper.

- The current Introduction and Context sections will be merged to provide clearer context for this paper and a clarified mandate.

- The context and mandate will be clarified by first exposing the EA's current objectives to transition to probabilistic flood forecasts (from recent policy documents).

- The pitfalls to meet these objectives will be clearly identified in this new merged Introduction-Context section.

- We will conclude the Introduction-Context section by stating how this work aims to tackle some of the pitfalls identified, through these interviews and paper. This will form our clearly reformulated and contextualised research question.

- This will form a clearer storyline leading up to a solid research question and interview results, analysis and recommendations.
- This section will also contain literature review information about the need for probabilistic flood forecasts in practice and the challenges of decision-making facing uncertainty. This is to expand the context of this work to the wider geoscience communication theme.

3) Similarly, the section 4 is very interesting; it is neat to have a window into the ways of working of these flood forecasters. However, the quotes and text do not form a coherent story for the reader. The section 4 does not seem well linked to the recommendations in section 5 – it is not clear to the reader how the interviews resulted in the recommendations. They read more like a policy brief than a research output; perhaps this would be better suited as a commentary than a research article?

Using quotes to support textual insights is a very common format for interview-based publications and constitutes a valid research article format. We will explain this paper format beforehand in the Methods section, supported by similar paper formats in the field. We do however appreciate that some readers may be unfamiliar with this format and will clarify links between the text and quotes where needed in the paper.

In order to link more clearly the interview results and recommendations, we will:

- Link Table 1 topics with 1 or 2 recommendations.
- Rewrite the Discussion section to combine Sections 5.1 and 5.2, where each paragraph will present: interview finding(s) – literature finding(s) – 1 or 2 recommendations.

4) In particular, section 5 seems to bring in the idea of decision-makers, but all of the interviews were with forecasters and those responsible for issuing warnings. There was no analysis in section 4 of who the decision-makers are, what decisions would be made based on the warning, or how that affected the information that was released.

The duty officers interviewed are decision-makers at the heart of the forecasting chain. They collaboratively decide whether to issue flood warnings operationally and pass relevant information on to other decision-makers further down the chain (both internal and external to the EA; e.g. the public, flood incident duty officers and emergency responders). When discussing Fig. 1, we will clarify the duty officers' roles as decision-makers within the EA. We will also highlight and briefly describe other decision-makers in the chain, internal and external to the EA, who act on the information provided by the duty officers.

5) The recommendations in section 5 seem to be generally good practice recommendations for whenever a forecasting system might have some sort of change, but not necessarily linked to the research results. In addition, most of the recommendations do not seem to have any relation with the characteristics of this particular change in forecast systems; the fact that the new system is probabilistic rather than deterministic.

See response to comment 3) above. Together with the EA co-authors of this paper, we will make these recommendations more specific and clearer.

6) Here are some examples of additional places where the text could be changed to present a clear research statement to the reader: Section 3.3 question 2 – as a reader first encountering this

statement, I am not sure what you mean by this. Section 3.3 question 3 – when you say "potential impacts", what does this mean? As a reader, I can't anticipate what exactly you are looking for.

Question 2 will be merged with Question 1, both in the question wording and Results section. Indeed, the variety of information used by the duty officers highlighted in Section 4.2 (corresponding to Question 2) is an integral part of the duty officers' roles.

By "potential impacts" we mean the challenges and opportunities that this transition might lead to for MFDOs and FWDOs. The wording of Question 3 will be clarified.

7) The text would benefit from a few additional examples, if possible.

A few additional relevant examples will be added to the text, with supporting quotes, to guide readers more clearly through the results storyline.

8) Some additional points for clarification: Page 6 line 213: How is running the NFFS different from the information given by the FFC? How will those NFFS localized model runs update or change the flood forecast of the FFC? It would be helpful to have a worked example, of what was produced by the FFC, what was communicated to the forecasters, what additional data they gathered, and what warnings they then communicated to people and what decisions were made based on that information.

The NFFS contains locally tailored hydrological and hydraulic models, which provide local flood forecast information, complementing the national and county scale information provided by the FFC, as mentioned on Page 6 lines 202-204. This will be clarified in the paragraph on Page 6 lines 212-219.

The decision-making storyline/worked example you mention seems a very good approach to narrating how duty officers make decisions. We will rewrite parts of the current text in Section 4.1 to reflect this new narrative, supported (if possible) by EA graphics.

9) Page 6 line 234: What do you mean that they wait for the forecast to be "confident"? Is this not a deterministic forecast?

As stated in the glossary of technical terms (Appendix A), confident here refers to "A forecaster's expert judgement of how certain they are that the forecast is right". The forecast is composed of two scenarios, which might sometimes show very diverging outcomes. This may lead to the duty officer being less "confident" about the signal shown by the forecast and the decision to make. Furthermore, combining different sources of information (highlighted in Section 4.1.1, but also in Section 4.2; e.g. national/county scale forecasts, mode performance information, river level correlations), the FWDO will add some expert judgment to gauge whether they can "trust" what the two forecast scenarios show. This will be clarified both in the text and in the glossary. This should make more sense once Sections 4.1 and 4.2 are merged (see response to comment 6) above).

10) The title is eye-catching, but the question that is being asked by the interviewee (water over bank vs. Armageddon) is about the magnitude of the event rather than about probabilities, while the paper is about transitioning to probabilistic forecasts.

We think that the title captures the paper's content very adequately. Indeed, the question raised by one of the interviewees and used in the title: "Are we talking just a bit of water out of bank? Or is it

Armageddon?" reflects the binary perspective of duty officers on this decision-making problem. This is a challenge at the heart of this paper and at the heart of the probabilistic forecast-lead decision-making process. The title will be explained in the paper abstract and introduction.

---

## Author Comment (AC3) · 17 Dec 2019

**Authors' responses to interactive comment RC2 from Anonymous Referee #2**

Dear authors,

Thank you for the interesting read. To me this is a valuable insight in the reality of flood forecasting and (early) warning in practice, provided first hand, by the duty officers of the UK Environment Agency. I appreciate the choice of the authors to present many quotes from the interviews performed, which helps in their attempt to make an authentic account of the current practice, as well as the expectations of the upcoming introduction of a probabilistic flood forecasting system.

We thank the reviewer for their positive comments with regards to this paper's added value in published literature and format, as well as their valuable feedback which will help improve the paper for final publication.

I have the following general comments:

1) The experience with the recent (2013/2014) transition from single forecast to 2-scenarios may be quite relevant for the perception of the interviewees and prospects of the upcoming change from two scenarios to probabilistic forecasts. Did you discuss this with the interviewees and could you perhaps elaborate more on this aspect in the paper?

We did not discuss this explicitly during interviews as not all interviewees had experienced this transition. However, a few interviewees compared the single forecast to the two scenarios and overall seemed positive about the added value of the scenarios. This reinforces the need for this paper to be published. Indeed, if similar work/interviews had been done for the previous transition to the two scenarios there could have been written records of the challenges and opportunities it presented to help the current transition to probabilistic forecasts.

2) From reading the paper I get the impression that there is little known yet about what will be the procedures for preparing, communicating internally, and use in warning decision making of the probabilistic forecasts. Could you, for example in the Context section, elaborate on what is known, and what was known to the interviewees at the time of the interviews, about the upcoming transition to probabilistic forecasts? If nothing is known yet, the good thing is that the recommendation of co-design (recommendation 3) can still be taken up, and at the same time it might explain some of the perceived challenges associated with the upcoming transition to using probabilistic forecasts.

This is a very good point, which we will elaborate on in the Introduction-Context Section. There was indeed very little known about the communication and forecast use procedures and no internal procedures in place at the time of the interviews. This motivated this work and interviews. Regarding the interviews, while some interviewees knew about the transition and were involved in the technical design of the new forecasting system (as part of their daily job at the EA, when not on duty), a few interviewees had just learnt about the transition a few days/hours prior to the interviews.

3) The presentation and discussion of the answers of the interviewees to the last question, about the upcoming introduction of probabilistic flood forecasts, seems to me to be somewhat limited. This impression is fed by the sudden change in reporting format from in-line quotes to a wordcloud,

summarising table (Table 1), and a reference to an Appendix, none of which are discussed in the manuscript text (lines 460-464). I may be overlooking something, and if not, you may well have chosen this approach for good reasons. If by design, then I would recommend to explain the reasons in the same section (Section 4.3.2). If possible, however, I would recommend continuing with the reporting format of the previous sections, or at least including a discussion of the wordclouds and Table 1.

We fully agree with the reviewer on this point. We had chosen to summarise this information in a table and wordclouds in order to present results succinctly and to shorten the overall length of the paper. However, in hindsight, we agree that this is one of the main highlights of the paper and needs more elaboration. We will incorporate this section into the same format as the other result sections, with a mix of text and supporting quotes.

4) Also the Discussion and Recommendation section leaves me with a feeling that more reflection on the interview results can be done. It would be, for example, interesting to reflect on whether the interviewees' answers to the first questions are in-line with, help explain, or not, their perceived opportunities and challenges (last question). The 10 recommendations in section 5.2 seem somewhat disconnected from the interview results (only Refs to literature are given). I would recommend to put in, in Section 5, more references to findings reported in earlier sections of the paper.

In order to link more clearly the interview results and recommendations, we will:

- Link Table 1 topics with 1 or 2 recommendations.
- Rewrite Discussion section to combine Sections 5.1 and 5.2, where each paragraph will present: interview finding – literature finding – recommendation.

Specific questions and comments:

5) You focus on the benefit of probabilistic forecasts of increasing lead time (e.g. p1 l14 and l36, p3/4 l124/125, while other benefits include the potential of increasing the probability of detection of floods (reducing missed events), and supporting risk-based decision making. Could you reflect on this in the text? E.g. adding advantages or explaining why you refer mainly to increasing lead time. Interview findings indicate that in the current practice final decision of issuing a warning is often based on nowcasts with lead times of only a couple of hours and/or on observations (e.g. page 7 line 267 and page 11 line 397/398). Do the interviewees and/or you think that the introduction of probabilistic forecasts is going to change this practice? Could you reflect on this in the paper?

This is a very good point which we have overlooked and not explicitly mentioned here. We will add these additional benefits of probabilistic forecasts to the paper.

The authors' hope is indeed that this transition to probabilistic forecasts will be reflected in the EA's decision-making practice (e.g. the lead time at which warnings are issued). This is reflected in Recommendation 7 (Page 7 starting line 590). We will develop this point further in the Discussion section. It was not explicitly mentioned by any of the interviewees.

6) Some of the duty officers (DOs) seem to be concerned about how the probabilistic forecasts will be received by the action response units. Does this mean that in the current practice, forecast hydrographs are send along with the warning to action response units? Please clarify in the text.

To our knowledge the information shared with emergency responders is only textual. Probabilistic information would however also affect verbal communication, which some duty officers indeed expressed worries about. This will be clarified in the text.

7) I may have missed it, but could you include more information (and refs if available) on the probabilistic forecasting system that will be used? Is it based on meteorological ensembles or on another probabilistic forecast method? Will the MFDOs be responsible of running it through the hydrological models (as they seem to be now)? Will hydrological uncertainty also be included in the hydrometeorological ensemble, and how? etc. Please also reflect on whether this information on the features of the new forecasting system was known to the interviewees.

This was not known at the time of the interviews. Together with the EA co-authors of this paper, we will try to provide some more information about the new probabilistic forecasting system, if possible.

8) When reading Table 1, I do not perceive a strong concern about the upcoming introduction of probabilistic forecasts, while when reading the quotes of Annex C I do sense a strong concern among the Duty Officers interviewed. This concern seems mainly to be that probabilistic forecasts will put all the responsibility of taking a decision with themselves (rather than with the forecasters or with the action response units). Could you note and discuss this in Section 4.3.2.? And then elaborate on recommendations on how to prevent/manage that? For example, Recommendation 9), setting guidelines on '..the forecast confidence at which certain decisions and actions should be made..', may also not be the answer, because the DOs indicated in the present-day practice the value of local expert judgement in issuing warnings and seem to appreciate having the freedom of applying such expertise. Prescribing decision making rules, may, therefore, be a step too far in taking away forecast interpretation responsibility from them.

We will reword the content of Table 1 and reformat it as per comment 3) to capture the quotes more adequately.

We understand and agree to an extent with the reviewer's comment about taking away forecast interpretation responsibility from the duty officers. We however believe that this could perhaps be a starting point (i.e. recommended decision), from which duty officers could be allowed to deviate when needed. This will be discussed further with the EA paper co-authors in order to propose an adapted (set of) recommendation(s) to tackle this.

9) Could you reflect on whether the interviewees see the probabilistic forecasts as an additional input to their flood forecast confidence assessment? The DOs are already communicating a confidence level with the warnings they send-on down the line. One quote in Appendix 3 confirms seeing this as an opportunity, but it would be interesting to read from you what is your impression on this for the other interviewees.

A few interviewees mentioned the fact that probabilistic forecasts would reveal uncertainty otherwise hidden with the flood scenarios, as is indeed reflected in Quote O2, and some words of

the left wordcloud (Fig. 5; e.g. "apparent", "displays", "reveal", "hidden", etc.). This will be expanded on in the text.

10) Page 4 line 130/131 refers to the EA already using probabilistic coastal flood forecasts. Why not learn from the experiences of that earlier transition (perhaps too long ago), or at least from the user experiences. Could you elaborate a bit more, e.g. whether or not you think that would be interesting for other researchers and the EA to pick up.

This was not explored during these interviews as not all interviewees were familiar with coastal flood forecasts (given the non-proximity of some of the EA centres where interviews were carried out to the coast). A few lines will be added to the Discussion section, suggesting this as future research to learn from this past transition to coastal flood probabilistic forecasts for the current transition for fluvial floods.

11) Page 5 lines 182-184: Consider referring back to these research questions in your Conclusions section.

This will be addressed.

12) Page 6 lines 222-227: Choosing the What-if scenario could be perceived as quite a responsibility. A responsibility that might be (partly) taken away with the introduction of probabilistic forecasts. Did you discuss this with the MFDOs and what are their and your thoughts on this? Consider elaborating on this in the paper.

This was indeed mentioned in Table 1, under "The forecasting system": "Some interviewees mentioned that the two scenarios, and the What If scenarios used to produce them, were sometimes challenging to play with and required a lot of expert judgment, thus making them inconsistent nation-wide." It will be expanded on when the table is adapted into text (as per comment 3) above).

13) Are all the warnings issued being archived (including alerts, issue time, updates, etc.)? Are actual flood occurrences being documented? And if so, are the archives being compared and analysed? Could you reflect on this, and do you think such analysis could be/has been helpful for identifying challenges in the current forecasting system and warning practices, as well as for analysing in the near future the impact (or lack thereof) when introducing the probabilistic forecasts and identifying persistent and potentially new challenges?

The 'Flood Intelligence Files' compile information (e.g. highest events on record, what rainfall led to them, what the catchment state was at the time and any known impacts) for every gauge the EA is providing forecasts for. Further information (e.g. whether the warnings are being logged as well for post-event analysis) will be added to the paper after discussions with the EA paper co-authors. The use of such a system to monitor the transition's performance in practice will be elaborated on in the Discussion section. If currently non-existent, such a system would be very valuable indeed and will be added to the recommendations.

14) Page 11 line 391/392: Could these differences perhaps also be a consequence of differences in catchment size/rainfall-runoff response time/land use and differences in flood management actions

that follow the warnings and the time these measures take? Consider mentioning/reflecting on this (at this point) in the paper.

This is indeed true and the historical differences are also caused by catchment response differences as per our interview discussions, and possibly by the amount of time it takes to prepare in anticipation for a flood (partly controlled by the catchment size too). This will be mentioned in the text.

15) Page 11 line 400: Are the DOs being scored, and if so, how are they scored, and what are these scores used for? If possible, would be interesting to comment on below this citation, and perhaps consider to reflect on how such scoring may have an encouraging or discouraging impact on the uptake of probabilistic forecasts.

This could be a figure of speech used by the interviewee and will be explored further with the EA paper co-authors. It will be clarified in this section and subsequently discussed in the Discussion section, alongside discussion about FWDOs' worries of the transition to probabilistic forecasts moving "the burden of making a decision further down the tree" (Page 14 line 539).

16) The paper concerns the upcoming transition from a 2-scenario forecast to a probabilistic forecast, but the Supplementary material seems to focus on the recently completed transition from a single forecast to the two scenario's (following 2013/2014). Please clarify, e.g. in the author response, not necessarily in the manuscript.

The Supplementary material indeed displays examples of the current system (two scenario-based), as the future probabilistic system is still in the making. This will be clarified in the paper in the Appendix caption.

Detailed comments and editorials:

We will make the following changes

p1 l12 Consider ..inclusion of uncertainty information in..

p1 l13 Consider ..potential upcoming floods.. instead of 'future' to avoid confusion with climate change. Also consider for other occurrences of 'future'.

p1 l18 Consider ..understand their perception on how this transition..

p1 l24 This sentence is rather broad and in my view not necessary. Consider leaving it out. Instead consider putting some of the key findings and recommendations (similarly to what is written in the Conclusions section)

We will leave this sentence in the abstract as it is important to highlight at this stage of the paper that a glossary of terms is available, as this is a paper for Geoscience Communication and should hence be able to communicate these findings for readers from a range of disciplines.

We will however include a few more specific key findings to the Abstract, and thank the reviewer for this suggestion.

p1 l38/39 Consider ..given the explicit provision of uncertainty information.. Single forecasts are as uncertain as probabilistic forecasts. The uncertainty is just not shown.

p2 l42/43 ..designed to capture scenarios that may not always realise.. That does not sound quite right/the point of probabilistic forecasting. Consider just leaving that whole sentence out, or reformulate to something like "Warning on the basis of low probabilities of flood, for example, will reduce the chance of missing an event, but will also lead to more false alarms."

p2 l44 Consider .. when a pre-defined threshold (e.g. river stage) is reached..

p3 l88 Consider ..whilst local flood authorities..

p3 l113 I would suggest including here an explanation on how the two scenarios are prepared. (I realise this is described later, but it left me curious from this point onward, especially because it matters for the context of the interviews to what extend these two scenarios are a step towards probabilistic forecasting or not).

This will be addressed and clarified at this stage when the Introduction and Context sections are merged.

p3 l114-118 I am not sure if I now understand correctly how the two scenarios should be used. This might be due to me not being a native speaker, but if you could reformulate to further clarify, that would be appreciated.

This is vocabulary used by the EA around these scenarios and cannot be paraphrased. We will however expand on this further (as per the comment above) to clarify this point.

p3 l120 ..potential risks of impacts.. Please reformulate.

This will be rephrased to: "Allowing to quantify the potential impacts of upcoming floods and their associated likelihood"

p4 l129 I do not think you can 'ensure' the appropriate use. Consider to reformulate, e.g. 'support'.

p5 l171 ..advance..

p5 l186 ..communicated by several interviewees,..

p6 l234 'waiting for the forecast to be confident' Please explain how the forecast can become confident in the context referred to here (present practice).

As stated in the glossary of technical terms (Appendix A), "confident" here refers to "A forecaster's expert judgement of how certain they are that the forecast is right". The forecast is composed of two scenarios, which might sometimes show very diverging outcomes. This may lead to the duty officer being less "confident" about the signal shown by the forecast and the decision to make. Furthermore, combining different sources of information (highlighted in Section 4.1.1, but also in Section 4.2; e.g. national/county scale forecasts, mode performance information, river level correlations), the FWDO will add some expert judgment to gauge whether they can "trust" what the two forecast scenarios show. This will be clarified both in the text and in the glossary.

p7 l257 please add who has the 'Expert knowledge'.

The FWDOs' expert knowledge and the knowledge contained in the 'Flood Intelligence Files' (see Page 10 lines 373-374). This will be clarified here.

p7 l264 please also describe how/for what/when the 'reasonable worst case' should be used.

It is "used for preparation, information and response to flooding" (Page 3 line 116). As stated on Page 9 lines 343-345: "For certain types of events, such as convective rainfall events*, for which the duty officers know models are still limited, they might decide to issue a warning based on the 'Reasonable Worst Case', although it is "technically against procedure" [MFDO2]." Section 4.2 will be merged with Section 4.1 and this will thus become clearer at this stage (as per responses to RC1).

p9 l345 please consider to add also the procedure for using the reasonable worst case again.

See comment above.

P10 l380 - 383 seem to me a bit too personal. Kindly double-check.

We believe that it is a valid and important perspective to quote. The interviewee will however be contacted to verify that they are fine with this quote being in the paper.

p12 l453 In my view this is an important finding that can also be used in the upcoming transition (and gives reason for a positive prospect). I cannot recall whether you clearly refer back to this finding in the Discussion and Conclusion sections at the end of the paper, but if not, I recommend including it.

We absolutely agree and will mention it again and discuss it in the Discussion section.

p13 l463/464 Unclear, and appears as a stand-alone sentence. I suggest making this part of a discussion to be added, of Fig. 5, Table 1, and Appendix C. (See also my third general comment)

We agree - see our answer to comment 3) above.

p13 l463 consider ..sound extreme.. or alternative formulation.

We will reformulate to: "sound extreme"

p13 l469/470 I do not understand this sentence. If we achieve increased confidence levels before moving up lead time, why is the second part of the sentence, about the chaotic system, posing problems? Consider clarifying or leaving out the sentence.

This will be rephrased to: "However, despite improvements in flood forecasting at increasing lead times, the predictability is still inherently limited by the chaotic nature of the system we are trying to model."

p13 l472/473 ..'uncertain' science.. Not sure I understand. Do you mean new discoveries being at first 'uncertain' (or not trusted), until the experiment has been reproduced with the same results (or tested in pilots, practice, etc.)? Or do you mean, more specifically, the science of quantifying uncertainty associated with predictions? Consider reformulating.

We refer here to the latter, the apparent uncertainty displayed by probabilistic predictions. We will rephrase to: "of new and probabilistic science".

p13 l490/491 consider ..decision makers operate, and where the forecast..

p13 l497 consider ..uncertainty information.. or ..information on uncertainty..

p14 l512/513 ..crucial to develop a methodology.. Not sure if a single methodology is the 'crucial' solution of the challenge of using probabilistic forecasts. It may be that case-dependent and user-community dependent ways have to be found, by scientists and users together, on how to effectively use confidence (uncertainty) information.

We agree and will rephrase this.

p14 l518 I do not think this reflects the main idea of probabilistic forecasts, nor that the provision of scenarios causes more false alarms. I would argue the other way around, that having the scenarios, the probabilistic forecasts, gives the opportunity to the decision maker (and its beneficiaries) to balance the number of missed events and false alarms to their needs. Please reformulate.

We will reformulate to reflect the main point of this paragraph: "A transition to probabilistic flood forecasts should be reflected in an institution's wider flood management priorities."

p14 l525 This seems a bit out of the blue and not clear to what extend you think this relates specifically to EA communication pathways and warning procedures.

This entire paragraph will be reformulated to capture the main point better (as per the comment above).

p14 l531 Consider adding a brief explanation on what is a 'post-factual society'.

p15 l547 consider ..we made a list..

We used the present tense here as these recommendations are relevant and actionable now.

p15 l548 consider something like ..The recommendations concern actions we think the EA should take with high priority..

p15 recommendation 1: Before this campaign, should there first be known a bit more on how the change will be done, or not?

We agree to an extent, but think that the EA shouldn't wait for the entire system to be set up before starting to tell people about the changes to come. This will give key actors a chance to be involved for a more successful transition. This will be reflected in the text.

p15 recommendation 2: consider ..to all players..

p15 l577 is this recommendation specifically for EA, or for the flood forecasting and early warning research community. If the latter, consider moving this elsewhere.

This recommendation is both general and specific to the EA. This will be clarified and make more sense when the discussion is merged (see response to comment 4) above).

p16 l582/583 given the reported differences in catchments, forecast performance, impacts, and warning response actions, etc., double-check this recommendation. Consider 'customised' rather than 'homogeneous'.

We like the word "customised" as it reflects inherent differences amongst centres and areas (as stated in the paper).

p16 l601 consider ..should be collected and used to update design and procedures..

p16 l602 consider ..To handle situations..

p16 l613 stand-alone sentence, consider moving somewhere else or elaborating.

p17 l623 because of the mentioned concern of the responsibility of decision making being pushed down to the DOs, I would not write 'lie mostly outside their role', rather just something like 'the main perceived challenges concern..'. In the sentence before, perhaps it would be more clear to write something like '..concerns about impacts of this transition on the communication and interaction between them.'

We agree with this point and will rephrase this sentence.

p17 l624 consider replacing 'translating uncertain information to a binary decision' (because it is a challenge that they already perceive in the present situation) for the worry of the responsibility of decision making being pushed further to them.

p17 l625 consider reformulating to something like ..High priority actions were recommended to the EA.. to support a successful..

p23 Caption figure 4: consider ..Complex flood forecast interpretation landscape.. because the decision making landscape includes more elements, such as external pressure.

p23 Caption figure 5: please add from what the opportunities and challenges arise (e.g. ..from the introduction of probabilistic flood forecasts)

p24 Title table 1: consider ..A sample of supporting quotes..

This will be adapted into text and the comments below will be incorporated.

p24 Table 1 line 2: 'improve long-term communication' please clarify.

p24 Table 1 line 2: add a full-stop at the end of the sentence.

p24 Table 1 line 5: consider ..contain information on forecast uncertainty..

p24 Table 1 line 22: ..it is worth noting that..

---

## Author Comment (AC4) · 17 Dec 2019

**Authors' responses to interactive comment RC3 from Jan Verkade**

Overall impression

The manuscript describes the 'transition' of the Environment Agency towards using probabilistic fluvial flood forecasts. The topic is very relevant and worthy of analysis indeed.

We thank the reviewer, Jan Verkade, for his very constructive feedback, which will help improve this manuscript for its final publication.

Having said that, I think the manuscript would benefit from (i) a better description of what it aims to achieve (ii) additional analysis that would justify the recommendations that are made.

Re (i). The research, through the interviews, provides data that documents the view of various individuals at a single point in time, regarding a change in an organization's processes and procedures. I like to make the analogy with "observations" in the quantitative sciences. Observations can be used to provide evidence of the plausibility of some hypothesis - or of the absence thereof. Such a 'hypothesis' element would greatly improve the quality of the manuscript - and provide a response to the "So What?" question that, post reading the manuscript, continues to linger in my mind.

We will rewrite the Introduction and Context, which, combined, will provide a better framing for this paper. This new Introduction section will highlight the EA's current policy and objectives with regards to undertaking/implementing probabilistic forecasts in practice. This will form a basis for the "hypothesis" element and a framework for the objectives of this paper. We will additionally clarify why these interviews were undertaken and why they are important to help capture and document this significant transition. Because this paper should be of wider benefit to the geoscience communication community, we will also answer the "So what?" question more clearly in the Discussion, highlighting what we have learnt from these interviews for other geoscience communication situations.

Re (ii). Based on the interview data, recommendations are phrased. While some of these may be very worthwhile indeed, I think recommendations can only be made based on an analysis where objectives are offset with achievements or projected achievements. Ergo, I think recommendations can only be made if the agency's objectives (with respect to the production and use of probabilistic forecasts) are described. Have these objectives been described in the 2016 National Flood Resilience Review that is cited, maybe? Or are documented elsewhere?

This is partly answered in the response to the comment above. We will rewrite the Introduction and Context into one Section which lays out the landscape in which this work finds itself. We will refer to policy documents about the forthcoming transition, as well as about the transition to 2 scenarios, as both of these transitions are part of a wider move.

Maybe the above issues could be resolved through the following:

We thank Jan Verkade for this step by step help into improving the paper's context and rationale. We will adapt the Introduction and Context sections, as well as the Discussion, as per the comments below.

- There is a document that says that the EA should be moving towards probabilistic forecasting. (NFRR but also the 2008 Pitt review?)
  And the report by Dale et al. (2013). We will expand on statements and findings from these documents (and others) in the paper.
- This overall objective has been adopted by the agency and existing projects/policies (evidence, please!) are in place to try and achieve that objective. As an aside, I wonder if this is indeed the case.
  While these are internal documents, we will work with the EA paper co-authors to find citeable EA current and future practice documents to make this context more tangible in the Introduction.
- Specifically, this means that EA will have to do this-and-this ('specific objectives'). If these specific objectives have been described in a policy document, great - use these in your manuscript. If not described, make them up - what could be plausible objectives?
  We will tackle this point to clarify the context and objectives of this paper in the Introduction. We will provide evidence for the UK government's and the EA's policy to move towards probabilities. While the new probabilistic flood forecasting system is currently being developed at the EA, there is lack of clarity about how this will affect the decision-makers, who are key players in the system and will ensure that it is successful in practice. This is the rationale for our paper.
- To meet those specific objectives, the agency will have to do this-and-this. This would be the description of your organizational transition.
  See response to comment above.
- We've gathered some data to try and identify where in that process the EA currently is, what pitfalls they see and where they think the challenges are.
  One of the pitfalls which we will identify is the lack of clarity about how this will affect the decision-makers, who are key players in the system and will ensure that it is successful in practice. This is the rationale for our paper (see response above).
- Offsetting specific objectives versus these 'observations', we note that . . . agency is well on its way / straying from its path . . . either way, recommendations are. . .
  These interviews provide the basis for the recommendations we make, which will be rephrased to link more clearly with the interview results/"observations". We will however not be able to comment on the EA's overall progress on the transition, as this depends on many other elements which we are not tackling in this paper. We will clarify this in the paper.
- It's useful to publish this is in the scientific literature so that (i) scientists may comment on implementation of science in a public organizations; (ii) other organizations may benefit from this; (iii) in assessing the progress of their transition, the EA can benefit from this analysis.
  And (iv) useful for the geoscience community, within the wider topic of communicating complex science for decision-making within operational organisations.

So overall - I think this manuscript is a rough diamond that needs polishing. I am very much interested in seeing the end result - and would be happy to help out through additional reviews if these would be considered helpful.

Other points

I think the above would require fairly significant restructuring of the manuscript and I don't think it's worthwhile to, at this stage in the review process, point out any minor issues.

Some scattered observations though:

- The citation in the title does not pertain to the theme of the manuscript. I also find that the manuscript tends to use language that, at times, can be a little more 'dramatic' than required. This is exemplified by the title's "front line perspectives" (a change in processes and procedures is not a war!) and the reference to "Armageddon" (which, by the way, is a settlement on top of a hill - the 'Ar' originates from the Hebrew 'har' which means mountain - and not prone to flooding and hence reference thereto is somewhat unfortunate - but I am digressing now). Additional examples: "the chaotic and far from certain world we live in", "urgently required", "high priority recommendations".
  We will tone down some of the language. However, where language has been used in quotes or there is precedent in the current policy in this area we will leave it. For example, "front line" is language often used by the EA. We think that the title captures the paper's content very adequately. Indeed, the question raised by the interview and used in the title: "Are we talking just a bit of water out of bank? Or is it Armageddon?" reflects the binary perspective of duty officers on this decision-making problem. This is a challenge at the heart of this paper and at the heart of the probabilistic forecast-lead decision-making process. The title will be explained in the paper abstract and introduction.
- A glossary is, I think, unnecessary, and I find the asterisks a little distracting.
  We disagree and think this is very helpful for readers outside of this field of expertise, who are interested about geoscience communication.
- The manuscript's theme is the 'transition to probabilistic forecasting'. The amount of text that is dedicated to that theme, however, is relatively small. For example, the Results sections spans lines 190 through 464 - yet only as of line 425 are the probabilistic forecasts discussed. Similar observations can be made to the manuscript as a whole. In my view, the reader is distracted a lot from the main points.
  We agree and will restructure the paper to highlight its main points more clearly. Section 4.2 will be merged with 4.1. Section 4.3 (highlight result section) will be reformatted into the same format as the other result sections, with a mix of text and supporting quotes. Table 1, Fig. 5 and Appendix C will all be expanded on in the text. In order to link more clearly the interview results and recommendations, we will:
  - Link Table 1 topics with 1 or 2 recommendations.
  - Rewrite Discussion section to combine Sections 5.1 and 5.2, where each paragraph will present: interview finding – literature finding – recommendation.
- The language used to describe the somewhat technical aspects of predictive uncertainty could be a little more precise. Some examples:
  - It's not forecasts themselves that are uncertain. What's uncertain is the future water levels, streamflows, etc. - also even if an estimate of those future values is made through a forecast. When that residual uncertainty is quantified or expressed, we have available 'estimates of uncertainty', rather than 'uncertainty'.

> We will make sure that this is clarified throughout the paper: the forecasts display the uncertainty in our estimates of the future water levels, streamflows, etc.

- o Uncertainty estimates and probabilistic forecasts are not the same thing. Hopefully the level of uncertainty can be expressed as a probability, but very often it cannot. I wouldn't want to use the terms interchangeably in a manuscript.

  > We will be careful to rephrase relevant misleading parts of the paper and will clarify this in the Glossary.

- o In a manuscript that discussed 'uncertainty', you have to be a little careful with using the word 'certain' ('certain decisions', l609). In most of those cases, the word 'certain' can be either safely omitted, or replaced by 'various'.

  > We agree and will find synonyms.

- Minor, minor issue: In author contributions, why not simply reference first names? "Hannah, Louise and Susan posed the original question" is a lot easier to read than "H.L.C., L.An. and S.M. posed the original question".

  > This is the convention and we will keep this author contributions format.

---

## Author Response (AR1)

Dear Editor and Reviewers,

Please find below our point by point response (blue) to the reviewers' comments (black). Comments from several reviewers regarding a similar point/issue (each new bullet point) were combined for clarity. Note that the page and line numbers indicated in our response refer to the manuscript document with edits displayed.

- **RC1:** The title is eye-catching, but the question that is being asked by the interviewee (water over bank vs. Armageddon) is about the magnitude of the event rather than about probabilities, while the paper is about transitioning to probabilistic forecasts.
  **RC3:** The citation in the title does not pertain to the theme of the manuscript.

  We think that the title captures the paper's content very adequately. Indeed, the question raised by one of the interviewees and used in the title: "Are we talking just a bit of water out of bank? Or is it Armageddon?" reflects the binary perspective of duty officers on this decision-making problem. This is a challenge at the heart of this paper and at the heart of the probabilistic forecast-lead decision-making process. This was clarified and tied in with paper results in Section 3 (P12 L425).

- **RC1:** While the concept of the paper is excellent, my major suggestion is to re-work the framing. As a reader, it is difficult to follow the logic of the paper. Often, it seems to be an interesting mix of quotes from interviews, lacking some analysis to identify relevant points and present them in a structured manner.

  We have restructured the paper as a whole. The Introduction was merged with the former Context section to provide a succinct yet thorough overview of the topic at heart and a clearer framing for the content of the paper (now Section 1). Section 4 was restructured to provide a storyline account of the current EA practice (now Section 3), emphasised by the numbered arrows added to Figure 2. Former Section 4.3 was merged with former Section 5 in a more thorough analysis and discussion of what might be the future EA practice given the transition (now Section 4). See responses below for more in-depth explanations.

- **RC1:** The introduction provides interesting context to the flood forecasting situation in the UK, but the details often seem disconnected. It is not quite clear how the authors are leading up to a solid research question. At the end of the introduction, it becomes clear that the EA is going to transition to probabilistic forecasts, and that the researchers will interview the forecasters about this transition. However, it is not really clear what the purpose is for the interviews. My suggestion is to re-write the introduction with a clarified mandate, leading up to the question that will be then answered in the paper.
  **RC3:** Re (i). The research, through the interviews, provides data that documents the view of various individuals at a single point in time, regarding a change in an organization's processes and procedures. I like to make the analogy with "observations" in the quantitative

sciences. Observations can be used to provide evidence of the plausibility of some hypothesis - or of the absence thereof. Such a 'hypothesis' element would greatly improve the quality of the manuscript - and provide a response to the "So What?" question that, post reading the manuscript, continues to linger in my mind. Re (ii). Based on the interview data, recommendations are phrased. While some of these may be very worthwhile indeed, I think recommendations can only be made based on an analysis where objectives are offset with achievements or projected achievements. Ergo, I think recommendations can only be made if the agency's objectives (with respect to the production and use of probabilistic forecasts) are described. Have these objectives been described in the 2016 National Flood Resilience Review that is cited, maybe? Or are documented elsewhere?

We have rewritten the Introduction and merged it with the former Context section to clarify the context and purpose of this paper, with a hypothesis on P6 L224-226.

- **RC1:** Similarly, the section 4 is very interesting; it is neat to have a window into the ways of working of these flood forecasters. However, the quotes and text do not form a coherent story for the reader.

  We appreciate that some readers may be unfamiliar with this format and have reorganised and rewritten parts of former Section 4 to make a more coherent story for the reader, following the chain of information Duty Officers lead ahead of a potential flood event (now Section 3). We have also numbered the quotes for cross-referencing.

- **RC1:** The section 4 does not seem well linked to the recommendations in section 5 – it is not clear to the reader how the interviews resulted in the recommendations. They read more like a policy brief than a research output; perhaps this would be better suited as a commentary than a research article? The recommendations in section 5 seem to be generally good practice recommendations for whenever a forecasting system might have some sort of change, but not necessarily linked to the research results. In addition, most of the recommendations do not seem to have any relation with the characteristics of this particular change in forecast systems; the fact that the new system is probabilistic rather than deterministic.
  **RC2:** The presentation and discussion of the answers of the interviewees to the last question, about the upcoming introduction of probabilistic flood forecasts, seems to me to be somewhat limited. This impression is fed by the sudden change in reporting format from in-line quotes to a wordcloud, summarising table (Table 1), and a reference to an Appendix, none of which are discussed in the manuscript text (lines 460-464). I may be overlooking something, and if not, you may well have chosen this approach for good reasons. If by design, then I would recommend to explain the reasons in the same section (Section 4.3.2). If possible, however, I would recommend continuing with the reporting format of the previous sections, or at least including a discussion of the wordclouds and Table 1. Also the Discussion and Recommendation section leaves me with a feeling that more reflection on the interview results can be done. It would be, for example, interesting to reflect on whether

the interviewees' answers to the first questions are in-line with, help explain, or not, their perceived opportunities and challenges (last question). The 10 recommendations in section 5.2 seem somewhat disconnected from the interview results (only Refs to literature are given). I would recommend to put in, in Section 5, more references to findings reported in earlier sections of the paper.

**RC3:** The manuscript's theme is the 'transition to probabilistic forecasting'. The amount of text that is dedicated to that theme, however, is relatively small. For example, the Results sections spans lines 190 through 464 - yet only as of line 425 are the probabilistic forecasts discussed. Similar observations can be made to the manuscript as a whole. In my view, the reader is distracted a lot from the main points.

**SC1:** Given the quotes listed in the appendix, I think the wording of table 1 could be stronger.

Former Sections 4.3 and 5 were merged and rewritten to provide a more thorough analysis and discussion of what might be the future EA practice given the transition (now Section 4), replacing the previous Table 1. We have used the same reporting format (text supported by quotes) as the current Section 3, as recommended by RC2. Section 4 is now structured in 5 sub-sections to reflect the structure and provide a better reflection of the current practice (Section 3). Each sub-section provides: several quotes to communicate the interviewees perceptions of how a particular topic may be affected by the transition, an analysis of these quotes and the current practice (cross-referencing quotes from Section 3, which are now numbered), literature findings on the given topic, which combined should logically lead to a clear and specific recommendation for the EA. We have rewritten and clarified the recommendations to reflect the findings of this paper.

- **RC1:** In particular, section 5 seems to bring in the idea of decision-makers, but all of the interviews were with forecasters and those responsible for issuing warnings. There was no analysis in section 4 of who the decision-makers are, what decisions would be made based on the warning, or how that affected the information that was released.

  **SC1:** Largely the paper considers the FWDOs as the decision makers as they are the ones issuing flood warnings. The latter part of the paper also introduced external stakeholders as decision makers, making decisions on the EA flood warnings. It would improve clarity to make it clearer when you are talking about internal and external decision makings. In particular this applies to the discussion around lines 500 and 540. Maybe also consider adding a definition of 'decision maker' to the glossary and recognising that decisions are made at various points during the forecasting chain. Similarly it is not clear who makes the decisions re increasing control room staffing or increasing field resources (e.g. for putting up flood defences). Is this the FWDO, the MFDO or someone else? And what time scale are these decisions made on/how will the change to probabilistic forecasts affect them?

  The Duty Officers interviewed are decision-makers at the heart of the forecasting chain. They collaboratively decide whether to issue flood warnings operationally and pass relevant information on to other decision-makers further down the chain (both internal and external

to the EA; e.g. the public, flood incident duty officers and emergency responders). We have clarified their roles as decision-makers (and the roles of other decision-makers in the chain) in Section 3.1 (P8 L300-323). At the end of this section, we have also clarified that we refer to the Duty Officers when talking about decision-makers throughout the paper, unless stated otherwise. We have also added a definition of 'decision-maker' to the Glossary (P48). The timescales of response were clarified throughout Section 3 (e.g. P8 L327-328, P9 L338-350) and reflected on in Section 4 with regards to the transition to probabilistic forecast (P31-32 L1040-1059).

- **RC1:** Here are some examples of additional places where the text could be changed to present a clear research statement to the reader: Section 3.3 question 2 – as a reader first encountering this statement, I am not sure what you mean by this. Section 3.3 question 3 – when you say "potential impacts", what does this mean? As a reader, I can't anticipate what exactly you are looking for.
  **RC2:** Page 5 lines 182-184: Consider referring back to these research questions in your Conclusions section.

  The research questions were reworded into two aims which provide the structure for the results of this paper (Section 2, P7 L286-290). The aims were discussed in the Conclusions.

- **RC1:** The text would benefit from a few additional examples, if possible.

  Sections 3 and 4 were rewritten to provide a clearer storyline, and screenshots of FFC and EA products were added for clearer examples of the incident response process to support the text (Appendix B).

- **RC1:** Some additional points for clarification: Page 6 line 213: How is running the NFFS different from the information given by the FFC? How will those NFFS localized model runs update or change the flood forecast of the FFC? It would be helpful to have a worked example, of what was produced by the FFC, what was communicated to the forecasters, what additional data they gathered, and what warnings they then communicated to people and what decisions were made based on that information.

  The NFFS contains locally tailored hydrological and hydraulic models, which provide local flood forecast information, complementing the national and county scale information provided by the FFC, as previously mentioned on P6 L202-204. This was clarified throughout Section 3.2.2. We have rewritten Section 3 to provide a storyline of how Duty Officers process information and make decisions ahead of a potential flood event, supported by additional visual examples in Appendix B. However, as these visual examples were not available for a specific flood event (the EA does not keep all past forecast products, as added on P18 L623-620, "This database however does not capture catchment conditions or

forecasts produced at the time the warning was issued") we had to keep the text fairly general, while constructing a clearer storyline.

- **RC1:** Page 6 line 234: What do you mean that they wait for the forecast to be "confident"? Is this not a deterministic forecast?
  **RC2:** p6 l234 'waiting for the forecast to be confident' Please explain how the forecast can become confident in the context referred to here (present practice).

  As previously stated in the Glossary, "confident" here refers to "A forecaster's expert judgement of how certain they are that the forecast is right". The forecast is composed of two scenarios, which might sometimes show very diverging outcomes. This may lead to the duty officer being less "confident" about the signal shown by the forecast and the decision to make. Furthermore, combining different sources of information (previously highlighted in Section 4.1.1, but also in Section 4.2; e.g. national/county scale forecasts, mode performance information, river level correlations), the FWDO will add some expert judgment to gauge whether they can "trust" what the two forecast scenarios show. The use of the term 'confident' was clarified in Section 3 (P11 L402-403) and in the Glossary under 'confident' (P48).

- **RC2:** The experience with the recent (2013/2014) transition from single forecast to 2-scenarios may be quite relevant for the perception of the interviewees and prospects of the upcoming change from two scenarios to probabilistic forecasts. Did you discuss this with the interviewees and could you perhaps elaborate more on this aspect in the paper?

  We did not discuss this explicitly during interviews as not all interviewees had experienced this transition. However, the breadth of the interviewees' responses could hint at the interviewees' experience with this past transition. This merits to be explored further. We have added a paragraph to highlight this in Section 4 (P36 L1253-1257).

- **RC2:** From reading the paper I get the impression that there is little known yet about what will be the procedures for preparing, communicating internally, and use in warning decision making of the probabilistic forecasts. Could you, for example in the Context section, elaborate on what is known, and what was known to the interviewees at the time of the interviews, about the upcoming transition to probabilistic forecasts? If nothing is known yet, the good thing is that the recommendation of co-design (recommendation 3) can still be taken up, and at the same time it might explain some of the perceived challenges associated with the upcoming transition to using probabilistic forecasts. I may have missed it, but could you include more information (and refs if available) on the probabilistic forecasting system that will be used? Is it based on meteorological ensembles or on another probabilistic forecast method? Will the MFDOs be responsible of running it through the hydrological models (as they seem to be now)? Will hydrological uncertainty also be included in the

hydrometeorological ensemble, and how? etc. Please also reflect on whether this information on the features of the new forecasting system was known to the interviewees.

What was (un)known about the transition to probabilistic forecasting at the time of the interviews was clarified in the Introduction (P5 L206-210) and in Section 4 (P25 L846-848), with supporting references for the readers to find further information (as this does not fall within the scopes of this paper).

- **RC2:** You focus on the benefit of probabilistic forecasts of increasing lead time (e.g. p1 l14 and l36, p3/4 l124/125, while other benefits include the potential of increasing the probability of detection of floods (reducing missed events), and supporting risk-based decision making. Could you reflect on this in the text? E.g. adding advantages or explaining why you refer mainly to increasing lead time.

  The benefits of probabilistic forecasting were clarified in the Introduction (P4-5 L166-178).

- **RC2:** Interview findings indicate that in the current practice final decision of issuing a warning is often based on nowcasts with lead times of only a couple of hours and/or on observations (e.g. page 7 line 267 and page 11 line 397/398). Do the interviewees and/or you think that the introduction of probabilistic forecasts is going to change this practice? Could you reflect on this in the paper?

  We indeed think that the introduction to probabilistic forecast should change this practice and reflected on how the lead times of response based on probabilistic forecasts might change from the current practice in Section 4 (P31-32 L1040-1059).

- **RC2:** Some of the duty officers (DOs) seem to be concerned about how the probabilistic forecasts will be received by the action response units. Does this mean that in the current practice, forecast hydrographs are send along with the warning to action response units? Please clarify in the text.

  What was known to us and shared by the EA about the exchange with emergency responders was written in Section 3 (P8 L317-320).

- **RC2:** When reading Table 1, I do not perceive a strong concern about the upcoming introduction of probabilistic forecasts, while when reading the quotes of Annex C I do sense a strong concern among the Duty Officers interviewed. This concern seems mainly to be that probabilistic forecasts will put all the responsibility of taking a decision with themselves (rather than with the forecasters or with the action response units). Could you note and

discuss this in Section 4.3.2.? And then elaborate on recommendations on how to prevent/manage that? For example, Recommendation 9), setting guidelines on '..the forecast confidence at which certain decisions and actions should be made..', may also not be the answer, because the DOs indicated in the present-day practice the value of local expert judgement in issuing warnings and seem to appreciate having the freedom of applying such expertise. Prescribing decision making rules, may, therefore, be a step too far in taking away forecast interpretation responsibility from them.

Table 1 was adapted into text (see Section 4). We discussed the Duty Officers' concern about having more responsibility to interpret the probabilistic forecast to making a decision in Section 4 (P30-31 L1001-1029). At the end of this paragraph, we make a recommendation to tackle this.

- **RC2:** Could you reflect on whether the interviewees see the probabilistic forecasts as an additional input to their flood forecast confidence assessment? The DOs are already communicating a confidence level with the warnings they send-on down the line. One quote in Appendix 3 confirms seeing this as an opportunity, but it would be interesting to read from you what is your impression on this for the other interviewees.

The MFDOs indeed see probabilistic forecast as a potential source of additional confidence when communicating the forecast with FWDOs. This was explored further in Section 4 (P28 L931-960).

- **RC2:** Page 4 line 130/131 refers to the EA already using probabilistic coastal flood forecasts. Why not learn from the experiences of that earlier transition (perhaps too long ago), or at least from the user experiences. Could you elaborate a bit more, e.g. whether or not you think that would be interesting for other researchers and the EA to pick up.

This was not explored during these interviews as not all interviewees were familiar with coastal flood forecasts (given the non-proximity of some of the EA centres where interviews were carried out to the coast). However we think that the EA could take inspiration on the current coastal flood forecast decision-making to design probabilistic forecast-based decision-making rules. This was discussed in Section 4 (P31 L1022-1025).

- **RC2:** Page 6 lines 222-227: Choosing the What-if scenario could be perceived as quite a responsibility. A responsibility that might be (partly) taken away with the introduction of probabilistic forecasts. Did you discuss this with the MFDOs and what are their and your thoughts on this? Consider elaborating on this in the paper.

This was previously mentioned in Table 1, which has been adapted into text in Section 4 (P27 L901-903).

- **RC2:** Are all the warnings issued being archived (including alerts, issue time, updates, etc.)? Are actual flood occurrences being documented? And if so, are the archives being compared and analysed? Could you reflect on this, and do you think such analysis could be/has been helpful for identifying challenges in the current forecasting system and warning practices, as well as for analysing in the near future the impact (or lack thereof) when introducing the probabilistic forecasts and identifying persistent and potentially new challenges?

  The flood warnings are logged in a database, however not enough information is stored currently for post-event analyses. This was mentioned in Section 3 (P18 L621-625). We agree with the reviewer that this would indeed be very helpful in improving the current system and evaluating the impact of the transition to a new system. We discussed this and added a recommendation based on the discussion in Section 4 (P37 L1272-1279).

- **RC2:** Page 11 line 391/392: Could these differences perhaps also be a consequence of differences in catchment size/rainfall-runoff response time/land use and differences in flood management actions that follow the warnings and the time these measures take? Consider mentioning/reflecting on this (at this point) in the paper.

  This is indeed true and the historical differences are also caused by catchment response differences as per our interview discussions, and possibly by the amount of time it takes to prepare in anticipation for a flood (partly controlled by the catchment size too). This was mentioned in Section 3 (P18 L617-620).

- **RC2:** Page 11 line 400: Are the DOs being scored, and if so, how are they scored, and what are these scores used for? If possible, would be interesting to comment on below this citation, and perhaps consider to reflect on how such scoring may have an encouraging or discouraging impact on the uptake of probabilistic forecasts.

  This was a figure of speech and was clarified in Section 3 (P18 L621). It was additionally discussed in Section 4 (P30 L1009-1010), alongside discussion about FWDOs' concerns of having more responsibility in interpreting the probabilistic forecast to make a decision.

- **RC2:** The paper concerns the upcoming transition from a 2-scenario forecast to a probabilistic forecast, but the Supplementary material seems to focus on the recently completed transition from a single forecast to the two scenario's (following 2013/2014). Please clarify, e.g. in the author response, not necessarily in the manuscript.

The Supplementary material indeed displays examples of the current system (two scenario-based), as the future probabilistic system is still in the making. This was clarified in the paper in the Appendix B caption (P55).

- **RC2:** p1 l24 This sentence is rather broad and in my view not necessary. Consider leaving it out. Instead consider putting some of the key findings and recommendations (similarly to what is written in the Conclusions section)

  We have left this sentence in the abstract as it is important to highlight at this stage of the paper that a Glossary is available, as this is a paper for Geoscience Communication and should hence be able to communicate these findings for readers from a range of disciplines. We have however included a few more specific key findings to the Abstract (P1 L20-29).

- **RC2:** p3 l113 I would suggest including here an explanation on how the two scenarios are prepared. (I realise this is described later, but it left me curious from this point onward, especially because it matters for the context of the interviews to what extend these two scenarios are a step towards probabilistic forecasting or not). p3 l114-118 I am not sure if I now understand correctly how the two scenarios should be used. This might be due to me not being a native speaker, but if you could reformulate to further clarify, that would be appreciated.

  More and clearer information was provided about the two scenarios in the Introduction (P5 L194-203).

- **RC2:** p7 l257 please add who has the 'Expert knowledge'.

  This refers to the FWDOs' expert knowledge and the knowledge about potentially affected catchments contained in the 'Flood Intelligence Files'. This was rewritten in Section 3 (P14 L497-499 and 501-504).

- **RC2:** p7 l264 please also describe how/for what/when the 'reasonable worst case' should be used. p9 l345 please consider to add also the procedure for using the reasonable worst case again.

  The RWC should be used for incident planning, as clarified in Section 3 (P16 L559-566). However, in certain cases it may be used for warning, although technically against procedure, as mentioned in Section 3 (P18 L626-631).

- **RC2:** P10 l380 - 383 seem to me a bit too personal. Kindly double-check.

We have removed this quote as it was not of added value to the points made in the paper.

- **RC2:** p12 l453 In my view this is an important finding that can also be used in the upcoming transition (and gives reason for a positive prospect). I cannot recall whether you clearly refer back to this finding in the Discussion and Conclusion sections at the end of the paper, but if not, I recommend including it.

  We have discussed this in Section 4 (P25 L853-864) and mentioned it in the Conclusions (P37 L1296).

- **RC2:** p13 l463/464 Unclear, and appears as a stand-alone sentence. I suggest making this part of a discussion to be added, of Fig. 5, Table 1, and Appendix C. (See also my third general comment)

  We added further discussion in Section 4 (P32-36 L1087-1257) on how the interviewees' "resistance" to the transition to probabilistic forecast may hint at the lack of knowledge that the transition was happening, the wording of interview questions, and/or their experience with the previous transition to two scenarios. We also made recommendations based on these.

- **RC2:** p13 l469/470 I do not understand this sentence. If we achieve increased confidence levels before moving up lead time, why is the second part of the sentence, about the chaotic system, posing problems? Consider clarifying or leaving out the sentence.

  This sentence was removed.

- **RC2:** p13 l472/473 ..'uncertain' science.. Not sure I understand. Do you mean new discoveries being at first 'uncertain' (or not trusted), until the experiment has been reproduced with the same results (or tested in pilots, practice, etc.)? Or do you mean, more specifically, the science of quantifying uncertainty associated with predictions? Consider reformulating.

  We rephrased this to: "new and probabilistic science" and moved it to the Introduction (P1 L17).

- **RC2:** p15 recommendation 1: Before this campaign, should there first be known a bit more on how the change will be done, or not?

We agree to an extent but think that the EA shouldn't wait for the entire system to be set up before starting to tell people about the changes to come. As shown by some of the interviewees' responses, not knowing about a transition may cause personal resistance to the idea if not involved in shaping the transition. Making the transition public as it is happening will give key actors a chance to be involved for a more successful transition. This was clarified in Section 4's Recommendation 7 (P30 L993-1000) and expended on further (P32-36 L1087-1257).

- **RC2:** p14 l512/513 ..crucial to develop a methodology.. Not sure if a single methodology is the 'crucial' solution of the challenge of using probabilistic forecasts. It may be that case-dependent and user-community dependent ways have to be found, by scientists and users together, on how to effectively use confidence (uncertainty) information.

   We agree with the reviewer's comment, which is now reflected in Section 4's Recommendation 8 (P31 L1026-1029).

- **RC2:** p14 l518 I do not think this reflects the main idea of probabilistic forecasts, nor that the provision of scenarios causes more false alarms. I would argue the other way around, that having the scenarios, the probabilistic forecasts, gives the opportunity to the decision maker (and its beneficiaries) to balance the number of missed events and false alarms to their needs. Please reformulate. p14 l525 This seems a bit out of the blue and not clear to what extend you think this relates specifically to EA communication pathways and warning procedures.

   We have changed this paragraph to clarify its main point, that a transition to probabilistic flood forecasts should be reflected in an institution's wider flood management priorities. Based on the points raised in this paragraph an EA-specific recommendation was made (Section 4, P31-32 L1053-1059).

- **RC2:** p15 l577 is this recommendation specifically for EA, or for the flood forecasting and early warning research community. If the latter, consider moving this elsewhere.

   This recommendation was specific to the EA and clarified in Section 4 (P31 L1026-1029).

- **RC2:** p17 l623 because of the mentioned concern of the responsibility of decision making being pushed down to the DOs, I would not write 'lie mostly outside their role', rather just something like 'the main perceived challenges concern..'. In the sentence before, perhaps it would be more clear to write something like '..concerns about impacts of this transition on the communication and interaction between them.'

We agree with this point and will rephrase this sentence.

- **RC2:** p17 l624 consider replacing 'translating uncertain information to a binary decision' (because it is a challenge that they already perceive in the present situation) for the worry of the responsibility of decision making being pushed further to them.

  The entire Conclusions paragraph was rewritten.

- **RC3:** I also find that the manuscript tends to use language that, at times, can be a little more 'dramatic' than required. This is exemplified by the title's "front line perspectives" (a change in processes and procedures is not a war!) and the reference to "Armageddon" (which, by the way, is a settlement on top of a hill - the 'Ar' originates from the Hebrew 'har' which means mountain - and not prone to flooding and hence reference thereto is somewhat unfortunate - but I am digressing now). Additional examples: "the chaotic and far from certain world we live in", "urgently required", "high priority recommendations".

  We have toned down some of the language. However, where language has been used in quotes or there is precedent in the current policy in this area we have left it. For example, "front line" is language often used by the EA.

- **RC3:** The language used to describe the somewhat technical aspects of predictive uncertainty could be a little more precise. Some examples: It's not forecasts themselves that are uncertain. What's uncertain is the future water levels, streamflows, etc. - also even if an estimate of those future values is made through a forecast. When that residual uncertainty is quantified or expressed, we have available 'estimates of uncertainty', rather than 'uncertainty'. Uncertainty estimates and probabilistic forecasts are not the same thing. Hopefully the level of uncertainty can be expressed as a probability, but very often it cannot. I wouldn't want to use the terms interchangeably in a manuscript. In a manuscript that discussed 'uncertainty', you have to be a little careful with using the word 'certain' ('certain decisions', l609). In most of those cases, the word 'certain' can be either safely omitted, or replaced by 'various'.

  We have clarified throughout the paper with: the forecasts display the uncertainty in our estimates of the future water levels, streamflows, etc. (e.g. P5 L171-173). This was also explained in the Glossary under 'Uncertainty' and 'Probabilistic forecasts' (P50-51). We have also replaced the word 'certain' with synonyms.

- **SC1:** Firstly the importance of timings/lead time is confusing as the focus changes throughout the paper from the 2-6 hour window for issuing flood warnings to the 2-5 day window for more strategic planning decision making. The relative value, and expected use of probabilistic forecasts will be different at these two timescales. The quote on line 270 is key to this discussion. It would be helpful to have a clearer focus on what lead time is being discussed at different points in the paper. A specific example is line 234 talking about waiting for the forecast to be confident – I expect there is also a balance of confidence and lead time here

  We have clarified that the paper focuses on the window of 2 hours to 5 days ahead, as this is the timescale likely to be affected by this transition (Section 3, P8 L326-328).

- **SC1:** I know it is beyond the scope of the paper but there is limited consideration of how the change to probabilistic forecasts might affect external decision makers although it is raised as a concern by FWDOs. A useful extension to the work would be to see what information those receiving flood warnings actually want, to understand if it is a perceived or real concern about the lack of binary forecast information and the pushing of decision making further down the chain. This is also relevant to recommendation 1 – as well as communicating that there will be a change, should the EA also have some responsibility for preparing external decision makers on how this might change their practices?

  Although beyond the scope of this paper, we agree that this is indeed a very important point and that EA's strategic overview role includes preparing external decision-makers. We have included it in Section 4's Recommendation 7 (P30 L993-1000).

**Editorial changes**

- **RC2:** p1 l12 Consider ..inclusion of uncertainty information in..

  Rephrased (P1 L12): "By showing the uncertainty surrounding a prediction, probabilistic forecasts can give an earlier indication of potential upcoming floods, increasing the amount of time available to prepare."

- **RC2:** p1 l13 Consider ..potential upcoming floods.. instead of 'future' to avoid confusion with climate change. Also consider for other occurrences of 'future'.

  Done (P1 L13): see sentence above.

- **RC2:** p1 l18 Consider ..understand their perception on how this transition..

Rephrased (P1 L20): "to understand how they perceive this transition might impact on their decision-making."

- **RC2:** p1 l38/39 Consider ..given the explicit provision of uncertainty information.. Single forecasts are as uncertain as probabilistic forecasts. The uncertainty is just not shown.

  Rephrased (P5 L171-175 and P6 L211-212): "By indicating how likely a flood is to occur, probabilistic forecasts communicate an estimate of the uncertainty surrounding a prediction (expressed as a probability), and can support risk-based decision-making through an increased probability of detection of floods (reducing missed events*), an earlier indication of potential future extreme events, such as floods, and their associated impacts" and "Probabilistic forecasts can be challenging to use for operational decision-making*, given the explicit uncertainty information they communicate."

- **RC2:** p2 l42/43 ..designed to capture scenarios that may not always realise.. That does not sound quite right/the point of probabilistic forecasting. Consider just leaving that whole sentence out, or reformulate to something like "Warning on the basis of low probabilities of flood, for example, will reduce the chance of missing an event, but will also lead to more false alarms."

  Rephrased (P6 L215-216): "warning based on low probabilities of a flood, for example, will reduce the chance of missing an event, but might also lead to more false alarms."

- **RC2:** p2 l44 Consider .. when a pre-defined threshold (e.g. river stage) is reached..

  Rephrased (P6 L216-217): "Decisions can be made following a set of rules, such as threshold exceedance".

- **RC2:** p3 l88 Consider ..whilst local flood authorities..

  Removed as not crucial.

- **RC2:** p3 l120 ..potential risks of impacts.. Please reformulate.

  Rephrased (P5 L176-177): "by allowing to quantify the potential impacts of upcoming floods and their associated likelihood".

- **RC2:** p4 l129 I do not think you can 'ensure' the appropriate use. Consider to reformulate, e.g. 'support'.

  Changed to 'support' (P6 L231).

- **RC2:** p5 l171 ..advance..

  Done (P7 L277).

- **RC2:** p5 l186 ..communicated by several interviewees,..

  Done (P8 L295).

- **RC2:** p13 l463 consider ..sound extreme.. or alternative formulation.

  Sentence removed.

- **RC2:** p13 l490/491 consider ..decision makers operate, and where the forecast..

  Sentence removed.

- **RC2:** p13 l497 consider ..uncertainty information.. or ..information on uncertainty..

  Changed (P26 L874): "the design and communication of uncertainty information".

- **RC2:** p15 l547 consider ..we made a list..

  We kept the present tense as these recommendations are relevant and actionable now.

- **RC2:** p15 l548 consider something like ..The recommendations concern actions we think the EA should take with high priority..

  Rephrased to the sentence provided by the reviewer (P25 L841-842).

- **RC2:** p14 l531 Consider adding a brief explanation on what is a 'post-factual society'.

  Explanation added (P25 L983): "(i.e. culture in which public opinion depends on appeals to emotions rather than objective facts)".

- **RC2:** p15 recommendation 2: consider ..to all players..

  Recommendation removed.

- **RC2:** p16 l582/583 given the reported differences in catchments, forecast performance, impacts, and warning response actions, etc., double-check this recommendation. Consider 'customised' rather than 'homogeneous'.

  Used the word 'customised' (P27 L918): "we recommend that the EA carry out a locally tailored customised transition".

- **RC2:** p16 l601 consider ..should be collected and used to update design and procedures..

  Changed (P32 L1082-1083): "end-user feedback should be collected from all key players and considered to update the new system's design and procedures"

- **RC2:** p16 l602 consider ..To handle situations..

  Removed.

- **RC2:** p16 l613 stand-alone sentence, consider moving somewhere else or elaborating.

  Removed.

- **RC2:** p17 l625 consider reformulating to something like ..High priority actions were recommended to the EA.. to support a successful..

  Rephrased (P37 L1293-1294): "thirteen recommendations were spelled out to support a successful transition for flood early warning in England".

- **RC2:** p23 Caption figure 4: consider ..Complex flood forecast interpretation landscape.. because the decision making landscape includes more elements, such as external pressure.

  Changed (now Fig. 3): "Complex flood forecast interpretation landscape in which EA Duty Officers operate".

- **RC2:** p23 Caption figure 5: please add from what the opportunities and challenges arise (e.g. ..from the introduction of probabilistic flood forecasts)

  Figure removed as it did not add any additional value to the text.

- **RC2:** p24 Title table 1: consider ..A sample of supporting quotes.. line 2: 'improve long-term communication' please clarify. add a full-stop at the end of the sentence. line 5: consider ..contain information on forecast uncertainty.. line 22: ..it is worth noting that..

  This was adapted into text (Section 4).

- **RC3:** A glossary is, I think, unnecessary, and I find the asterisks a little distracting.

  We disagree and think this is very helpful for readers outside of this field of expertise, who are interested about geoscience communication. We have kept the Glossary and improved it following reviewers' comments.

- **RC3:** Minor, minor issue: In author contributions, why not simply reference first names? "Hannah, Louise and Susan posed the original question" is a lot easier to read than "H.L.C., L.An. and S.M. posed the original question".

  This is the convention and we have kept this author contributions format.

- **SC1:** Line 19 – I would remove 'alternative', I'm not sure why these two areas are alternative.

  Sentence removed.

- **SC1:** Lines 130:134 – this is largely a repeat of lines 76:81.

The former Introduction and Context sections were merged into a single Introduction, removing repetitions.

- **SC1:** Line 224 – increase/ change of 200%?

  Clarified (P10 L374-375): "(e.g. a 200% increase in catchment rainfall totals over the next 6 hours)".

- **SC1:** Personally figure 5 (the wordcloud) doesn't add much to the paper for me.

  We have removed this figure from the paper.

[revised manuscript text omitted]

---

## Referee Report (RR1)

**Worries and Views of the 'Decision Broker'**

The paper *"Are we talking just a bit of water out of bank? Or is it Armageddon?" Front line perspectives on transitioning to probabilistic fluvial flood forecasts in England* "by L. Arnal and co-authors give exciting insights into some operations of a flood warning system. The specific regional example and the circumstances of an upcoming change to the forecasting methodology illustrate the roles and attitudes of individuals who act as 'decision brokers'.

Compared to the initial version of the paper, the authors did rework the text substantially. The authors responded positively to numerous remarks of three referees. The revised text does not trigger the irritations that referees of the initial draft seem to have felt. If seen from that perspective, the text could be accepted. Nevertheless, in some cases, the authors could have aligned further with the views of the referees.

- Keeping the unchanged title looks like an unfortunate choice. Given the substance of the paper and the expectations of the reader, it would be better to alter the phrasing (for example: *Front line perspectives on transitioning to probabilistic fluvial flood forecasts in England – beyond: "Are we talking just a bit of water out of bank? Or is it Armageddon?"*).
- The reader would benefit from learning the purpose of the study at the beginning of the introduction. Shifting text from lines 110 -114 before line 33 would be a remedy.
- The notion "probabilistic science" seems to be a notion of limited explanatory power. Also, it has several meanings in different disciplines; as a literature search shows. How this notion is used in the abstract indicates further that the authors are less aware of studies of processes at the science-policy interface (e.g. McNie, Parris and Sarewitz, 2016; Kowarsch and Jabbour, 2017). This research puts in question the statement of the authors *"While science… the design of scientific practice"* (line 16-18).

Furthermore, the initial round of reviews did not emphasize some methodological limitations of the study.

- The sample of interviews is small; what is acknowledged by the authors.
- The chosen methodology is not critically reviewed. A minimal set of three bibliographic references is given, although several 10k publications using this methodology have been published since the most recent reference that the authors refer to; – see, for example, Kallio *et al.* (2016).
- The research of a local (England) and specific (fluvial flood forecasts) process is not embedded into studies of similar issues (e.g. shift of methodology for forecasting risks for the public) such as seismic risks or storm surges (e.g. Stewart and Lewis, 2017; Keith J Beven *et al.*, 2018; Keith J. Beven *et al.*, 2018).
- Numerous recommendations are made although they are based on a limited study (some hours of interviews). The authors should focus on some recommendations (for example, the first and third that are mentioned in the conclusions).

Notwithstanding these limitations, the unique subject of this study could justify the publication of the paper.

The interviews give a rare view into the 'engine room' of fluvial flood forecasts. Therefore, as the authors say (line 30), the subject of the study is of broad interest. The study describes a element at the intersection between geosciences and society, the 'decision broker' who transposes a scientific analysis (forecast) into a warning. Hopefully, there will be a follow-up to this limited study.

Beven, Keith J. *et al.* (2018) 'Epistemic uncertainties and natural hazard risk assessment – Part 1: A review of different natural hazard areas', *Natural Hazards and Earth System Sciences*, 18(10), pp. 2741–2768. doi: 10.5194/nhess-18-2741-2018.

Beven, Keith J *et al.* (2018) 'Epistemic uncertainties and natural hazard risk assessment – Part 2: What should constitute good practice?', *Natural Hazards and Earth System Sciences*, 18(10), pp. 2769–2783. doi: 10.5194/nhess-18-2769-2018.

Kallio, H. *et al.* (2016) 'Systematic methodological review: developing a framework for a qualitative semi-structured interview guide', *Journal of Advanced Nursing*, 72(12), pp. 2954–2965. doi: 10.1111/jan.13031.

Kowarsch, M. and Jabbour, J. (2017) 'Solution-oriented global environmental assessments: Opportunities and challenges', *Environmental Science and Policy*, 77(August), pp. 187–192. doi: 10.1016/j.envsci.2017.08.013.

McNie, E. C., Parris, A. and Sarewitz, D. (2016) 'Improving the public value of science: A typology to inform discussion, design and implementation of research', *Research Policy*. Elsevier B.V., 45(4), pp. 884–895. doi: 10.1016/j.respol.2016.01.004.

Stewart, I. S. and Lewis, D. (2017) 'Communicating contested geoscience to the public: Moving from "matters of fact" to "matters of concern"', *Earth-Science Reviews*. Elsevier, 174(February), pp. 122–133. doi: 10.1016/j.earscirev.2017.09.003.